# On Differentially Private Graph Sparsification and Applications

**Raman Arora**
Johns Hopkins University
`arora@cs.jhu.edu`

**Jalaj Upadhyay**
Rutgers University
`jalaj.kumar.upadhyay@gmail.com`

## Abstract

In this paper, we study private sparsification of graphs. In particular, we give an algorithm that given an input graph, returns a sparse graph which approximates the spectrum of the input graph while ensuring differential privacy. This allows one to solve many graph problems privately yet efficiently and accurately. This is exemplified with application of the proposed meta-algorithm to graph algorithms for privately answering cut-queries, as well as practical algorithms for computing MAX-CUT and SPARSEST-CUT with better accuracy than previously known. We also give an efficient private algorithm to learn Laplacian eigenmap on a graph.

## 1 Introduction

Data from social and communication networks have become a rich source to gain useful insights into the social, behavioral, and information sciences. Such data is naturally modeled as observations on a graph, and encodes rich, fine-grained, and structured information. At the same time, due to the seamless nature of data acquisition, often collected through personal devices, the individual information content in network data is often highly sensitive. This raises valid privacy concerns pertaining the analysis and release of such data. We address these concerns in this paper by presenting a novel algorithm that can be used to publish a succinct differentially private representation of network data with minimal degradation in accuracy for various graph related tasks.

There are several notions of differential privacy one can consider in the setting described above. Depending on privacy requirements, one can consider edge level privacy that renders two graphs that differ in a single edge as in-distinguishable based on the algorithm's output; this is the setting studied in many recent works [9, 19, 25, 54]. Alternatively, one can require node-level privacy which preserves privacy of each node, which has been the focus in [10, 11, 28, 44]. In this paper, we consider settings where nodes are known public entities and the edges represent sensitive or private events and attributes relating two nodes.

In particular, we consider the following notion of differential privacy. We say that an algorithm that takes a graph as input and returns a graph, is differentially private if given any two graphs that differ in a single edge by weight at most one [1], the output of the algorithm does not change by much (see Section 3 for a formal definition).

Given the ubiquitous nature of the problem, several recent works have studied various graph problems within the framework of differential privacy. These include the works on differentially private algorithms for computing degree distribution [28, 27, 44], on subgraph counting [40, 45], on private MIN-CUT [22] and on estimating parameters of generative graphical models [39]. Each of the works referenced above present algorithms that are tailor-made for a specific graph problems. We, on the

other hand, are interested in understanding the problem more generally. We pose the following question: given an input graph, can we efficiently generate a succinct yet private representation that allows us to simultaneously solve multiple graph-related tasks accurately?

Two popular representations of graphs that succinctly capture several graph properties include *spectral sparsification* and *cut sparsification*. Spectral sparsification [52] is the following problem: given a graph $\mathcal{G}$ with $n$ nodes, output a subgraph $\widetilde{\mathcal{G}}$ of $\mathcal{G}$ such that $(1 - \varepsilon)L_{\mathcal{G}} \preceq L_{\widetilde{\mathcal{G}}} \preceq (1 + \varepsilon)L_{\mathcal{G}}$, i.e.,

$$\forall x \in \mathbb{R}^n, \quad (1-\varepsilon)x^\top L_{\mathcal{G}} x \leq x^\top L_{\widetilde{\mathcal{G}}} x \leq (1+\varepsilon)x^\top L_{\mathcal{G}} x, \tag{1}$$

where $L_{\mathcal{G}}$ is the Laplacian of the graph $\mathcal{G}$ (see Definition 1). This is a generalization of cut sparsification [8], where $x$ is restricted to binary vector. Spectral sparsification of a graph is a fundamental problem that has found application in randomized linear algebra [13, 33], graph problems [29], linear programming [34], and mathematics [37, 51].

There are many non-private algorithms for computing spectral sparsification of graphs [2, 7, 35, 36, 50, 52]. However, to the best of our knowledge, there is no prior work on differentially private graph sparsification. This paper initiates this study by formally stating the goal of private graph sparsification and presents the first algorithm for efficiently computing a graph sparsification. The main contributions of this paper are as follows.

1. We show that differential privacy is not achievable under the traditional notion of spectral sparsification of graphs since the output itself may reveal information about the edges. Furthermore, we put forth an alternate but a well-posed formulation of differentially private graph sparsification problem.

2. We give an efficient algorithm that outputs a private sparse graph with $O(n/\varepsilon^2)$ edges. Since our output is a graph and preserves the spectrum of the input graph, we can solve many graph related combinatorial problems efficiently while preserving differential privacy.

The works most closely related to that of ours are that of Blocki et al. [9], Dwork et al. [19], and Gupta et al. [23]. Blocki et al. [9] and Dwork et al. [19] give an algorithm that returns a symmetric matrix that may not correspond to a graph but can be used to answer cut queries accurately while Gupta et al. [23] output a private graph that approximates cut functions, but cannot approximate spectral properties. Our algorithm for computing private sparse graph not only solves the two problems simultaneously but also improves upon both of these works.

Spectral sparsification of graphs finds many applications including, but not limited to, spectral clustering, heat kernel computations, separators, etc. Since differential privacy is preserved under post-processing, our result can be used in these applications to get a private algorithm for these tasks (see Table 1 in Section 4 for more details). On top of these improvements, we can leverage the fact that our output is a graph that approximates the spectrum of the input graph to efficiently compute *Laplacian eigenmap*, a useful technique used in manifold learning (see Section 4 for more details).

## 2   Preliminaries and Notations

The central object of interest in this paper is the Laplacian of a graph.

**Definition 1** (Laplacian of graph). Let $\mathcal{G}$ be an undirected graph with $n$ vertices, $m$ edges, and edge weights $w_e$. Consider an arbitrary orientation of edges of $\mathcal{G}$, and let $E_{\mathcal{G}}$ be the signed edge adjacency matrix of $\mathcal{G}$ given by

$$(E_{\mathcal{G}})_{e,v} := \begin{cases} +\sqrt{w_e} & \text{if } v \text{ is } e\text{'s head} \\ -\sqrt{w_e} & \text{if } v \text{ is } e\text{'s tail} \\ 0 & \text{otherwise} \end{cases}.$$

The *Laplacian* of $\mathcal{G}$ is defined as $L_{\mathcal{G}} = E_{\mathcal{G}}^\top E_{\mathcal{G}}$. Equivalently, $L_{\mathcal{G}} = D_{\mathcal{G}} - A_{\mathcal{G}}$, where $A_{\mathcal{G}}$ is the adjacency matrix and $D_{\mathcal{G}}$ is the degree matrix. One can verify that $L_{\mathcal{G}}\mathbf{1} = \mathbf{1}^\top L_{\mathcal{G}} = \mathbf{0}$, where $\mathbf{1}$ denotes all 1 vector and $\mathbf{0}$ denotes all 0 vector.

For a vertex $u \in V$, let $\delta_u \in \{0,1\}^n$ denote the row vector with 1 only at the $u$-th coordinate. Lets represent the edges of the graph with vectors $b_1, \ldots, b_m \in \{-1, 0, 1\}^n$ such that $b_e = \delta_u - \delta_v$ where the edge $e$ connects $u, v \in V$. Then, $L_{\mathcal{G}} = \sum_{e \in E_{\mathcal{G}}} b_e^\top b_e$. The spectral sparsification of a

graph can be casted as the following linear algebraic problem: given a sparsity parameter $s$ and row vectors $b_1, \ldots, b_m$, find a set of scalars $\tau_1, \ldots, \tau_m$ such that $|\{\tau_i : \tau_i \neq 0\}| \leq s$ and $(1-\varepsilon)L_\mathcal{G} \preceq \sum_{e \in E_\mathcal{G}} \tau_e b_e^\top b_e \preceq (1+\varepsilon)L_\mathcal{G}$. For any graph $\mathcal{G}$ with edge-adjacency matrix $E_\mathcal{G}$, the *effective resistance* (also known as leverage score) of the edge $e_i \in E_\mathcal{G}$ is defined as $\widetilde{\tau}_i := e_i^\top (E_\mathcal{G}^\top E_\mathcal{G})^\dagger e_i = e_i^\top L_\mathcal{G}^\dagger e_i$. It is well known that by sampling the edges (rows of $E_\mathcal{G}$) of $\mathcal{G}$ according to its leverage score, we obtain a graph $\widetilde{\mathcal{G}}$ such that $(1-\varepsilon)L_\mathcal{G} \preceq L_{\widetilde{\mathcal{G}}} \preceq (1+\varepsilon)L_\mathcal{G}$ (for example, see [50]).

**Notations.** We use the notation $(A \quad | \quad B)$ to denote the matrix formed by appending the columns of matrix $B$ to that of matrix $A$. We denote by $A^\dagger$ the Moore-Penrose pseudoinverse of $A$ and by $\|A\|_2$ its spectral norm. We use caligraphic letters to denote graphs, $V_\mathcal{G}$ to denote the vertices of $\mathcal{G}$ and $E_\mathcal{G}$ to denote the edges of $\mathcal{G}$. We drop the subscript when it is clear from the context. We use the symbol $K_n$ to denote an $n$ vertex complete graph and $L_n$ to denote its Laplacian. For any $S, T \subseteq V_\mathcal{G}$, the size of *cut* between $S$ and $T$, denoted by $\Phi_\mathcal{G}(S, T)$, is the sum of weight of the edges that are present between $S$ and $T$. When $T = V \backslash S$, we denote the size of cut between $S$ and $V \backslash S$ by $\Phi_\mathcal{G}(S)$. For a set $S \subseteq [n]$, we use the notation $\mathbf{1}_S = \sum_{i \in S} e_i$, where $\{e_1, \cdots, e_n\}$ denote the standard basis.

# 3 Differentially Private Graph Sparsification

We begin by noting that differential privacy is incompatible with the requirements of spectral sparsification in equation (1), because if we output a sparse subgraph of the input graph, then it will leak information about $\widetilde{O}(n)$ edges present in the graph. This motivates us to consider the following "relaxation" of the spectral sparsification problem.

**Definition 2** $((\varepsilon, \alpha, \beta, n)$-Private Spectral Sparsification)**.** Let $\mathfrak{G}$ be the set of all $n$ vertex positive weighted graphs. We are interested in designing an efficient algorithm $\mathcal{M} : \mathfrak{G} \to \mathfrak{G}$ such that

1. **(Privacy)** For all graphs $\mathcal{G}, \mathcal{G}' \in \mathfrak{G}$ that differ in only one edge by weight 1, and all possible measurable $S \subseteq \mathfrak{G}$, $\Pr[\mathcal{M}(\mathcal{G}) \in S] \leq e^\alpha \Pr[\mathcal{M}(\mathcal{G}') \in S] + \beta$.

2. **(Sparsity)** $\mathcal{M}(\mathcal{G})$ has at most $\widetilde{O}(n)$ edges, and

3. **(Spectral approximation)** $\widetilde{\mathcal{G}} \leftarrow \mathcal{M}(\mathcal{G})$ satisfies $(\kappa L_\mathcal{G} - \zeta L_n) \preceq L_{\widetilde{\mathcal{G}}} \preceq \eta L_\mathcal{G} + \xi L_n$ where functions $\eta, \xi, \kappa$ and $\zeta$ dependent on input parameters $n, \varepsilon, \alpha, \beta$.

The function $\zeta$ and $\xi$ can be seen as the distortion we are willing to accept to preserve privacy. That is, we would like $\zeta$ and $\xi$ to be as small as possible. Informally, we view adding $L_n$ as introducing plausible deniability pertaining to the presence of an edge in the output; it could be coming from either $\mathcal{G}$ or the complete graph.

The choice of $L_n$ is arbitrary and for simplicity. Our choice of $L_n$ is motivated by the fact that an $n$ vertex unweighted complete graph is the same irrespective of the input graph once the number of vertices is fixed. We can instead state item 3 by replacing $L_n$ by Laplacian of any graph whose edges are independent of any function of $\mathcal{G}$. For example, we can use a $d$-regular expander instead of $K_n$; however, this would not change our result as one can prove that a $d$-regular expander are spectral sparsification of $K_n$. In the definition, we consider *edge level privacy*, a setting studied in many recent works [9, 19, 25, 54][2].

We first enumerate why previous work do not suffice. The two works most related work to ours is by Blocki et al. [9] and Upadhyay [54] used random projection on the edge-adjacency matrix of graph $\mathcal{G}$ to output a matrix, $R$. One can show that the spectrum of $R^\top R$ approximates the spectrum of $L_\mathcal{G}$ if the dimension of random projection is high enough. It is claimed in Blocki et al. [9] that their output is a sanitized graph. However, even though their output is a Laplacian, one major drawback of random projection is that it does not preserve the structure of the matrix, which in our case is a edge-incidence matrix of a graph (we refer the readers to [13, 55] for more discussion on benefits and pitfalls of random projections). Likewise, the output of Dwork et al. [19] is a full rank matrix (and

hence not a Laplacian) that is not positive semi-definite matrix. Consequently, their result cannot be used for spectral sparsification and other applications considered in this paper. On the other hand, Gupta et al. [23] only approximates cut functions and not the spectrum.

The existing techniques for graph sparsification are either deterministic [7][3] or use importance sampling that depends on the graph itself [50, 52]. A popular approach (and the one we use) involves sampling each edge with probability proportional to its *effective resistance*. We show that effective resistance based sampling can be done privately, albeit not trivially.

In order to comprehend the issues with private sampling based on the effective resistance, consider two neighboring graphs, $\mathcal{G}$ and $\mathcal{G}'$, such that $\mathcal{G}$ has two connected components and $\mathcal{G}'$ has an edge $e$ between the two connected components of $\mathcal{G}$. No sparsifier for $\mathcal{G}$ will have the edge $e$; however, every sparsifier for $\mathcal{G}'$ has to contain the edge $e$. This allows one to easily differentiate the two cases. Furthermore, we show that the effective resistance is not Lipschitz smooth, so we cannot hope to privately compute effective resistance through output perturbation. One could argue that we can instead use (a variant of) *smooth sensitivity framework* [40], but it is not clear which function is a smooth bound on the sensitivity of effective resistance.

## 3.1 A High-level Overview of Our Algorithm

As we noted earlier, spectral sparsification of a graph can be posed as a linear algebra problem and computing effective resistance suffices for private sparsification of graphs. We propose an algorithm to sample using privately computed effective resistance of edges. Our algorithm is based on the following key ideas:

1. Private sketch of the left and right singular vectors of the edge-adjacency matrix is enough to compute all the effective resistances privately and a (possibly dense) private graph, $\mathcal{G}_{\mathsf{int}}$, that approximates the spectrum of the input graph.

2. We then sparsify $\mathcal{G}_{\mathsf{int}}$ to output a graph with $O(n/\varepsilon^2)$ edges with a small depreciation in the spectral approximation using any known non-private spectral sparsifier.

**Computing effective resistance privately.** We use the input perturbation technique (and its variants) first introduced by Blocki et al. [9] to compute effective resistances privately. The scalars $\{\tau_1, \ldots, \tau_m\}$ and $\mathcal{G}_{\mathsf{int}}$ are then computed using these effective resistances. Blocki et al. [9] first overlay a weighted complete graph $K_n$ on the input graph $\mathcal{G}$ to get a graph $\widehat{\mathcal{G}}$ and then multiply its edge-adjacency matrix $E_{\widehat{\mathcal{G}}}$ with a Gaussian matrix, say $N$. We view this algorithm as a private sketching algorithm, where the sketch is $R := N E_{\widehat{\mathcal{G}}}$. We prove in the supplementary material that if $N$ is of appropriate dimension as chosen in Step 2 of Algorithm 1, then $(1-\varepsilon)L_{\mathcal{G}} \preceq E_{\widehat{\mathcal{G}}}^{\top} N^{\top} N E_{\widehat{\mathcal{G}}} \preceq L_{\mathcal{G}}$. However, $R$ does not contains enough information to estimate the effective resistances of $E_{\widehat{\mathcal{G}}}$. For this, we sketch the left singular space as $L := \begin{pmatrix} E_{\widehat{\mathcal{G}}} & | & w\mathbb{I} \end{pmatrix} M$ for a Gaussian matrix $M$. We use different methods to sketch the right and left singular space so as to preserve differential privacy. Combined together $L$ and $R$ has enough statistics to privately estimate effective resistance, $\widetilde{\tau}_e$, of each edge $e$ using a simple linear algebra computation (Step 2 of Algorithm 1).

**Constructing $\mathcal{G}_{\mathsf{int}}$.** We overlay a weighted $K_n$ in order to privately sketch the left singular space leading to $\binom{n}{2}$ effective resistances. We define a set of scalars $\widetilde{\tau}_1, \ldots, \widetilde{\tau}_{\binom{n}{2}}$ using the computed effective resistance as in Step 2 of Algorithm 1. Let $\widetilde{\mathcal{G}}'$ be the graph formed by overlaying a weighted complete graph with edge weights sampled i.i.d. from a Gaussian distribution on $\widehat{\mathcal{G}}$. We cannot use existing non-private algorithms for spectral sparsification on graph $\widetilde{\mathcal{G}}'$ as it may have negative weight edges. To get a sanitized graph on which we can perform sparsification, we solve the following:

$$\text{SDP-1} : \min \left\{ \lambda : L_{\bar{\mathcal{G}}} - L_{\widetilde{\mathcal{G}}'} \preceq \lambda L_n, \ L_{\widetilde{\mathcal{G}}'} - L_{\bar{\mathcal{G}}} \preceq \lambda L_n, \ L_{\bar{\mathcal{G}}} \in \mathcal{C} \right\}.$$

Here $\mathcal{C}$ is the convex cone of the Laplacian of graphs with positive weights. Note that, SDP-1 has a solution with $\lambda = O(\frac{w}{n})$. This is because the original graph $\mathcal{G}$ achieves this value. Since, the semidefinite program requires to output a graph $\bar{\mathcal{G}}$ with Laplacian $L_{\bar{\mathcal{G}}}$, we are guaranteeed that its

**Algorithm 1** PRIVATE-SPARSIFY$(\mathcal{G}, \varepsilon, (\alpha, \beta))$

---

**Input:** An $n$ vertex graph $\mathcal{G} = (V_\mathcal{G}, E_\mathcal{G})$, privacy parameters: $(\alpha, \beta)$, sparsification parameter: $\varepsilon$.
**Output:** A graph $\widetilde{\mathcal{G}}$.

1: **Initialization.** Set $m = \binom{n}{2}, p = m + n$. Sample random Gaussian matrices $M \in \mathbb{R}^{p \times n}, N \in \mathbb{R}^{n/\varepsilon^2 \times m}, Q \in \mathbb{R}^{p \times p/\varepsilon^2}$, such that $M_{ij} \sim \mathcal{N}(0, \frac{1}{n}), N_{ij} \sim \mathcal{N}(0, \frac{\varepsilon^2}{n})$, and $Q_{ij} \sim \mathcal{N}(0, \frac{\varepsilon^2}{p})$.

2: **Set** $w = O(\frac{1}{\alpha \varepsilon} \sqrt{n \log n \log(1/\beta)} \log(1/\beta))$.

3: **Compute** $L := \left( E_{\widehat{\mathcal{G}}} \mid w\mathbb{I} \right) M, R := N E_{\widehat{\mathcal{G}}}$, where $E_{\widehat{\mathcal{G}}} = \sqrt{\left(1 - \frac{w}{n}\right)} E_\mathcal{G} + \sqrt{\frac{w}{n}} E_{K_n}$.

4: **Define** $\widetilde{\tau}_i = L_i \left( R^\top R \right)^\dagger L_i^\top$ for every $i \in [m]$ and $p_i = \min \left\{ c\widetilde{\tau}_i \varepsilon^{-2} \log(n/\delta), 1 \right\}$.

5: **Construct** a diagonal matrix $D \in \mathbb{R}^{\binom{n}{2} \times \binom{n}{2}}$ whose non-zero diagonal entries are $D_{ii} := p_i^{-1}$ with probability $p_i$.

6: **Construct** a complete graph $\mathcal{H}$ with edge weights i.i.d. sampled from $\mathcal{N}(0, \frac{12 \log(1/\beta)}{\alpha^2})$.

7: **Construct** $\widetilde{\mathcal{G}}'$ such that $L_{\widetilde{\mathcal{G}}'} = L_{\widehat{\mathcal{G}}} + L_{\mathcal{H}}$. Solve the SDP-1 to get a graph $\bar{\mathcal{G}}$.

8: **Output** $\widetilde{\mathcal{G}}$ formed by running the algorithm of Lee and Sun [35] on $E_{\widehat{\mathcal{G}}}^\top D E_{\bar{\mathcal{G}}}$.

---

output, $\bar{\mathcal{G}}$, will have $\lambda = O(\frac{w}{n})$. The Laplacian of the graph $\mathcal{G}_{\text{int}}$ is then constructed by computing $\sum_e \widetilde{\tau}_e u_e^\top u_e$, where $u_e$ are the edges of $\bar{\mathcal{G}}$. The graph $\mathcal{G}_{\text{int}}$ can be a dense graph, but we show that it approximates the spectrum of the input graph up to $\widetilde{O}(\frac{w}{n})$ additive error. To reduce the number of edges, we then run any existing non-private spectral sparsification algorithm to get the final output.

## 3.2 Main Result

We now state our result that achieves both the guarantees of Definition 2.

**Theorem 3** (Private spectral sparsification). *Given privacy parameter $(\alpha, \beta)$, accuracy parameter $\varepsilon$, an $n$ vertex graph $\mathcal{G}$, let $L_n$ denote the Laplacian of an unweighted complete graph, $K_n$, and $w = O\left( \frac{\log(1/\beta)}{\alpha \varepsilon} \sqrt{n \log n \log(1/\beta)} \right)$. Then we have the following:*

1. *PRIVATE-SPARSIFY is a polynomial time $(\alpha, \beta)$-differentially private algorithm.*

2. *$\widetilde{\mathcal{G}} \leftarrow$ PRIVATE-SPARSIFY$(\mathcal{G}, \varepsilon, (\alpha, \beta))$ has $O(n/\varepsilon^2)$ edges, such that with probability at least $9/10$, we have*

$$(1 - \varepsilon) \left( \left(1 - \frac{w}{n}\right) L_\mathcal{G} + \frac{w}{n} L_n \right) \preceq L_{\widetilde{\mathcal{G}}} \preceq (1 + \varepsilon) \left( \left(1 - \frac{w}{n}\right) L_\mathcal{G} + \frac{w}{n} L_n \right).$$

**Proof Sketch of Theorem 3.** Our algorithm requires performing matrix computations and solving a semi-definite program. All matrix computation requires at most $O(n^3)$ time. Solving a semi-definite program takes $\text{poly}(n)$ time, where the exact polynomial depends on whether we use interior point, ellipsoid method, or primal-dual approach. To prove privacy, we need to argue that computing $L$ and $R$ is $(\alpha/3, \beta/3)$-differentially private. Blocki et al. [9] showed that computing $R$ is $(\alpha/3, \beta/3)$-differentially private while computing $\mathcal{G}'$ is private due to Gaussian mechanism. The privacy guarantee on $L$ follows using arguments similar to [9].

For the accuracy proof, recall that we approximate the left singular space by $L$ and the right singular space by $R$ by using random Gaussian matrix, and then use $(L(R^\top R)^\dagger L^\top)_{ii}$ to approximate effective resistance $\widetilde{\tau}_i$ (see Step 2) – this follows from the concentration property of random Gaussian matrices. It is straightforward, then, to argue that sampling using the effective resistance thus computed would result in a graph. Proving that the estimated effective resistances provide the spectral approximation is a bit more involved and relies on matrix Bernstein inequality [53].

Our proof requires the spectral relation between $\bar{\mathcal{G}}$ and $\widehat{\mathcal{G}}$ since $X_i$ is defined with respect to the edges of $\bar{\mathcal{G}}$ and $\widetilde{\tau}_i$ is defined with respect to $\widehat{\mathcal{G}}$. Since the solution of SDP-1 is $\lambda = c\sqrt{\frac{\log n \log(1/\delta)}{n\alpha^2}}$, we have

$$L_{\widetilde{\mathcal{G}}'} - c\sqrt{\frac{\log n \log(1/\delta)}{n\alpha^2}} L_n \preceq L_{\bar{\mathcal{G}}} \preceq L_{\widetilde{\mathcal{G}}'} + c\sqrt{\frac{\log n \log(1/\delta)}{n\alpha^2}} L_n.$$

Another application of matrix Bernstein inequality gives us for some small constant $\rho > 0$,

$$\Pr\left[L_{\widehat{\mathcal{G}}} - \rho\sqrt{\frac{\log n \log(1/\delta)}{n\alpha^2}}L_n \preceq L_{\widetilde{\mathcal{G}}'} \preceq L_{\widehat{\mathcal{G}}} + \rho\sqrt{\frac{\log n \log(1/\delta)}{n\alpha^2}}L_n\right] \geq 1 - \delta.$$

Let $e_i$ be the edges in $\bar{\mathcal{G}}$ defined in Algorithm 1. Define random variables $X_i$ as follow:

$$X_i := \begin{cases} \frac{e_i e_i^\top}{p_i} & \text{with probability } p_i \\ 0 & \text{with probability } 1 - p_i \end{cases}.$$

Let $Y = \sum_i X_i$. Then we show the following:

$$\mathsf{E}[Y] = \sum_i \mathsf{E}[X_i] = L_{\bar{\mathcal{G}}} \quad \text{and} \quad X_i \preceq O\left(\frac{\varepsilon^2}{c\log(n/\delta)}\right)L_{\bar{\mathcal{G}}}.$$

Applying matrix Bernstein inequality [53] gives

$$\Pr\left[(1-\varepsilon)L_{\bar{\mathcal{G}}} \preceq Y \preceq (1+\varepsilon)L_{\bar{\mathcal{G}}}\right] \geq 1 - ne^{-c\varepsilon^{-2}\log(n/\delta)/3}$$

for large enough $c$. Now $E_{\bar{\mathcal{G}}}^\top D E_{\bar{\mathcal{G}}} = Y$ implies $(1-\varepsilon)L_{\bar{\mathcal{G}}} \preceq E_{\bar{\mathcal{G}}}^\top D E_{\bar{\mathcal{G}}} \preceq (1+\varepsilon)L_{\bar{\mathcal{G}}}$ with probability at least $99/100$. Now $E_{\bar{\mathcal{G}}}^\top D E_{\bar{\mathcal{G}}}$ can be the Laplacian of a dense graph. Using the result of Lee and Sun [35], we have $(1-\varepsilon)L_{\widetilde{\mathcal{G}}} \preceq L_{\widehat{\mathcal{G}}} \preceq (1+\varepsilon)L_{\widetilde{\mathcal{G}}}$ and that $L_{\widetilde{\mathcal{G}}}$ has $O(n/\varepsilon^2)$ edges. Combining all these partial orderings gives us the accuracy bound.

**Extension to Weighted Graphs** We can use a standard technique to extend the result in Theorem 3 to weighted graphs. We assume that the weights on the graph are integers in the range $[1, \text{poly } n]$. We consider different levels $(1+\varepsilon)^i$ for $i \in [c \log n]$ for some constant $c$. Then, we consider input graphs of the following form, $L_{\mathcal{G}} = \sum_{i=1}^{c \log n} L_{\mathcal{G},i}$, where $L_{\mathcal{G},i}$ has edges with weights $\{0, (1+\varepsilon)^i\}$. In other words, we use the $(1+\varepsilon)$-ary representation of weights on the edges and partition the graph accordingly and run Algorithm 1 on each $L_{\mathcal{G},i}$. Again, since there are at most $\text{poly} \log n$ levels, the number of edges in $\widetilde{\mathcal{G}}$ is $\widetilde{O}(\frac{n}{\varepsilon^2})$. Since $\widetilde{\mathcal{G}}$ is $(\alpha \, \text{poly} \log n, \beta \, \text{poly} \log n)$-differentially private, we can run another instance of [35] to get a graph $\widehat{\mathcal{G}}$ with $O(\frac{n}{\varepsilon^2})$ edges.

**Private Estimation of Effective Resistances.** Drineas and Mahoney showed a close relation between effective resistance and statistical leverage scores [16]. Their result imply that effective resistance are important statistical property of a graph. As a by-product of Algorithm 1, we can privately estimate the effective resitances of the edges. As mentioned earlier, this is not possible through previous approaches. For example, both Blocki et al. [9] and Dwork et al. [19] only approximates the right singular space, which is not sufficient to capture enough statistics to estimate the effective resistances. While effective resistance have been less explored, statistical leverage scores have been widely used by statisticians. Consequently over the period of time, researchers have explored some other statistical applications includes random feature learning [47] and quadrature [6] of effective resistances.

## 4  Applications of Theorem 3

Since differential privacy is preserved under any post-processing, the output of Algorithm 1 can be used in many graph problems. It is well known that the spectrum of a graph defines many structural and combinatorial properties of the graph [20]. Since Theorem 3 guarantees a sparse graph, it allows us to significantly improve the run-time for many graph algorithms, for example, min-cuts and separators [49], heat kernel computations [41], approximate Lipschitz learning on graphs [32], spectral clustering, linear programming [34], solving Laplacian systems [33], approximation algorithms [29, 42], and matrix polynomials in the graph Laplacian [12].

For example, Theorem 3 with Goemans and Williamson [21] (and Arora et al. [5]) allows us to output a partition of nodes that approximates MAX-CUT (SPARSEST-CUT, respectively). We can also answer all possible cut queries with the same accuracy as previous best in $O(|S|)$ time instead of $O(n|S|)$ time. This is a significant improvement for practical graphs. Lastly, we exhibit the versatility of our result by showing its application in extracting low-dimensional representations when data arise from sampling a probability distribution on a manifold, a typical task in representation learning.

| Problem | Additive Error | Previous Best Error | Theorem |
|---|---|---|---|
| $(S, V\backslash S)$ queries | $\widetilde{O}\left(\frac{|S|\sqrt{n}}{\alpha}\right)$ | $\widetilde{O}\left(\frac{|S|\sqrt{n}}{\alpha}\right)$ [9, 19] | Theorem 4 |
| $(S,T)$ queries | $\widetilde{O}\left(\frac{|S||T|}{\alpha\sqrt{n}}\right)$ | $\widetilde{O}\left(\frac{\sqrt{n}|S||T|}{\alpha}\right)$ [23] | Theorem 4 |
| MAX-CUT | $\widetilde{O}\left(\frac{|S_m|\sqrt{n}}{\alpha}\right)$ | $\widetilde{O}\left(\frac{n^{3/2}}{\alpha}\right)$ [23] | Theorem 5 |
| SPARSEST-CUT | $\widetilde{O}(\frac{1}{\sqrt{n}\alpha^2})$ | $\widetilde{O}(\frac{\sqrt{n}}{|S_s|\alpha})$ [23] | Theorem 5 |
| Eigenmap | $\widetilde{O}\left(\frac{\sqrt{n}}{\alpha}\right)$ | $\widetilde{O}\left(\frac{\sqrt{n}}{\alpha}\right)$ [19] | Theorem 6 |

Table 1: Applications of Our Results for $\beta = \Theta(n^{-\log n})$ ($\alpha > 0$ is an arbitrary constant, $S_m$ is vertex set inducing MAX-CUT, $S_c$ is vertex set inducing SPARSEST-CUT, $Q$ is vertex set for cut query).

In the rest of this section, we discuss these applications in more detail (see Table 1 for comparison). For this section, let $w = O(\frac{\log(1/\beta)}{\alpha\varepsilon}\sqrt{n\log n\log(1/\beta)})$.

**Answering $(S,T)$ cut queries efficiently.** Cut queries is one of the most widely studied problem in private graph analysis [9, 23, 54] and has wide applications [43, 46]. Given a graph $\mathcal{G}$, an $(S,T)$-cut query requires us to estimate the sum of weights of edges between the vertex sets in $S$ and $T$. Since our output is a graph, we can use it to answer $(S,T)$ cut queries privately and more efficiently.

**Theorem 4** (Cut queries). *Given a graph with $n$ vertices, there is an efficient $(\alpha, \beta)$-differentially private algorithm that outputs a graph $\widetilde{\mathcal{G}}$, which can be used to answer all possible $(S,T)$-cut queries with additive error $O(\frac{|S||T|}{\alpha\varepsilon}\sqrt{\frac{\log n\log^3(1/\beta)}{n}})$. In particular, when $T = V\backslash S$, then our algorihtm incur an additive error $O(\frac{\min\{|S|,|V\backslash S|\}}{\alpha\varepsilon}\log^3(1/\beta)\sqrt{n\log n\log(1/\beta)})$.*

This improves the result of Gupta et al. [23] who incur an $O(\frac{1}{\alpha}\sqrt{n|S||T|})$ additive error. Note that Blocki et al. [9] and Dwork et al. [19] can be used to only answer cut queries when $T = V\backslash S$. For $(S, V\backslash S)$-cut queries, recall that Dwork et al. output $C = E_{\mathcal{G}}^\top E_{\mathcal{G}} + N$, where $N$ is a Gaussian matrix with appropriate noise to preserve differential privacy. For a set of $q$ cut queries of form $(S, V\backslash S)$ with $|S| \leq |V\backslash S|$, standard concentration result of Gaussian distribution implies that Dwork et al. [19] incur $|\mathbf{1}_S^\top N\mathbf{1}_S| = O(|S|\sqrt{\log(1/\delta)\log(q)}/\alpha)$ additive error, where $\mathbf{1}_S \in \{0,1\}^n$ has 1 only in coordinate corresponding to $S \subseteq [n]$. In particular, if we wish to answer all possible cut queries, it leads to an additive error $O(|S|\sqrt{n\log(1/\beta)}/\alpha)$. We thus match these bounds [9, 19, 54] while answering an $(S, V\backslash S)$ query in $O(|S|/\varepsilon^2)$ amortized time instead of $O(|S|^2)$ amortized time.

**Optimization problems.** Given a graph $\mathcal{G} = (V, E)$ on a vertex set $V$, the goal of MAX-CUT to output a set of vertices $S \subseteq V$ that maximizes the value of $\Phi_S(\mathcal{G})$. It is well known that solving MAX-CUT exactly and even with $(0.87856 + \rho)$-approximation for $\rho > 0$ is NP-hard [31]. However, Goemans and Williamson [21] gave an elegant polynomial time algorithm for computing $0.87856 - \eta$ approximation to MAX-CUT for some $\eta > 0$, thereby giving an approximation algorithm that is optimal with respect to the multiplicative approximation. Another problem that is considered in graph theory is the problem of finding SPARSEST-CUT. Here, given a graph $\mathcal{G} = (V, E)$ on a vertex set $V$, the goal is to output a set of vertices $S \subseteq V$ that minimizes the value $\frac{\Phi_S(\mathcal{G})}{|S|(n-|S|)}$. The proposed algorithms for these problems first computes a private sparse graph as in Algorithm 1 followed by a run of the non-private algorithm ([21] in the case of MAX-CUT and [5] in the case of SPARSEST-CUT) to obtain a set of vertices $S$. We show the following guarantee.

**Theorem 5** (Optimization problems). *There is a polynomial-time algorithm that, for an $n$-vertex graph $\mathcal{G} := (V, E)$, is $(\alpha, \beta)$-differentially private with respect to the edge level privacy and produces a partition of nodes $(S, V\backslash S)$ satisfying*

$$\text{MAX-CUT:} \quad \Phi_S(\mathcal{G}) \geq (0.87856 - \eta)\left(\frac{1-\varepsilon}{1+\varepsilon}\right)\max_{S \subseteq V}\Phi_S(\mathcal{G}) - O\left(\frac{w|S|}{\alpha}\right).$$

*There is a polynomial-time algorithm that, for an $n$-vertex graph $\mathcal{G} := (V, E)$, is $(\alpha, \beta)$-differentially private with respect to the edge level privacy and produces a partition of nodes $(S, V\backslash S)$ satisfying*

$$\text{SPARSEST-CUT:} \quad \frac{\Phi_S(\mathcal{G})}{|S|(n-|S|)} \leq O(\sqrt{\log n})\left(\frac{1+\varepsilon}{1-\varepsilon}\right)\min_{S \subseteq V}\left(\frac{\Phi_S(\mathcal{G})}{|S|(n-|S|)}\right) + O\left(\frac{w\log^2 n}{n}\right).$$

The above theorem states that we can approximately and privately compute MAX-CUT and SPARSEST-CUT of an arbitrary graph in polynomial time. Further, the price of privacy in the form of the additive error scales sublinearly with $n$. On the other hand, if we use the privatized graph of [23], it would incur an error of $O(\frac{n^{3/2}}{\alpha})$ for MAX-CUT and $O(\frac{\sqrt{n}}{\varepsilon|S|})$ for SPARSEST-CUT.

One may argue that we can use the output of Dwork et al. [19] or Blocki et al. [9] to solve theses optimization problem. Unfortunately, it is not the case. To see this, let us recall their output. For a given graph $\mathcal{G}$, Blocki et al. [9] output $R^\top R$, where $R$ is as computed in Step 4 of Algorithm 1. On the other hand, Dwork et al. [19] computes $L_\mathcal{G} + N$, where $N$ is a symmetric Gaussian matrix with entries sampled i.i.d. with variance required to preserve privacy.

Both these approach output a symmetric matrix; however, the output of Dwork et al. [19] is neither a Laplacian nor a positive semi-definite matrix, a requirement in all the existing algorithms. On the other hand, even though the output of Blocki et al. [9] is a positive semi-definite matrix, it can have positive off-diagonal entries. As such, we cannot use them in the existing algorithms for MAX-CUT or SPARSEST-CUT since (analysis of) existing techniques for these optimization problems requires graph to be positively weighted (see the note by Har-Peled [24])[4]. Even if we can use their output, our algorithm allows a faster computation since we significantly reduce the number of constraints in the corresponding semi-definite programs for these optimization problems.

**Learning laplacian eigenmap.** A basic challenge in machine learning is to learn a low-dimensional representation of data drawn from a probability distribution on a manifold. This is also referred to as the problem of *manifold learning*. In recent years, many approaches have been proposed for manifold learning, including that of ISOMAP, local linear embedding, and Laplacian eigenmap.

In particular, the state-of-the-art Laplacian eigenmaps are relatively insensitive to outliers and noise due to locality-preserving character. In this approach, given $n$ data samples $\{x_1, \ldots, x_n\} \in \mathbb{R}^d$, we construct a weighted graph $\mathcal{G}$ with $n$ nodes and a set of edges connecting neighboring points. The embedding map is now provided by computing the top $k$ eigenvectors of the graph Laplacian. There are multiple ways in which we assign an edge $e = (u, v)$ and edge weight between nodes $u$ and $v$. We consider the following neighborhood graph: we have an edge $e = (u, v)$ if $\|x_u - x_v\|_2 \leq \rho$ for some parameter $\rho$. If there is an edge, then that edge is given a weight as per the *heat kernel*, $e^{-\|x_u - x_v\|_2^2/t}$ for some parameter $t \in \mathbb{R}$. The goal here is to find the embedding map, i.e., an orthonormal projection matrix, $UU^\top$, close to the optimal projection matrix, $U_k U_k^\top$, where the columns of $U_k$ are the top-$k$ eigenvectors of $L_\mathcal{G}$. Using our framework, we can guarantee the following for privately learning the Laplacian eigenmap matching the bound achieved by previous results [19].

**Theorem 6** (Laplacian eigenmap)**.** *Let $\mathcal{G}$ be the neighborhood graph formed by the data samples $\{x_1, \cdots, x_n\} \in \mathbb{R}^d$ as described above. Let $L_{\mathcal{G},k} = U_k U_k^\top L_\mathcal{G}$, where the columns of $U_k$ are the top-k eigenvectors of $L_\mathcal{G}$. Then, there is an efficient $(\alpha, \beta)$-differentially private learning algorithm that outputs a rank-k orthonormal matrix $U \in \mathbb{R}^{n \times k}$ such that*

$$\left\| (\mathbb{I} - UU^\top) L_\mathcal{G} \right\|_2 \leq (1 + \varepsilon) \left\| L_\mathcal{G} - L_{\mathcal{G},k} \right\|_2 + O(w).$$

## 5 Differentially Private Cut Sparsification

Benzur and Karger [8] introduced the notion of cut sparsification. In this section, we present an algorithm that outputs a cut sparsifier while preserving $(\alpha, 0)$-differential privacy. We use this algorithm to answer cut queries, approximately computing MAX-CUT, SPARSEST-CUT, and EDGE-EXPANSION, with $(\alpha, 0)$-differential privacy. We show the following:

**Theorem 7.** *Let $\mathcal{G} = (V, E)$ be an unweighted graph. Given an approximation parameter $0 < \varepsilon < 1$ and privacy parameter $\alpha$, PRIVATE-CUT-SPARSIFY, described in Algorithm 2, is $(\alpha, 0)$-differentially private. Further, PRIVATE-CUT-SPARSIFY outputs an $n$ vertices $\widetilde{O}(n/\varepsilon^2)$ edges graph, $\widetilde{\mathcal{G}}$, such that, with probability at least $99/100$, we have $\forall S \subseteq V$,*

$$(1-\varepsilon)\mathbf{1}_S^\top L_{\widetilde{\mathcal{G}}} \mathbf{1}_S - O\left(\sqrt{n\mathbf{1}_S^\top L_n \mathbf{1}_S}\right) \leq \alpha \cdot \mathbf{1}_S^\top L_\mathcal{G} \mathbf{1}_S \leq (1+\varepsilon)\mathbf{1}_S^\top L_{\widetilde{\mathcal{G}}} \mathbf{1}_S + O\left(\sqrt{n\mathbf{1}_S^\top L_n \mathbf{1}_S}\right).$$

**Algorithm 2** PRIVATE-CUT-SPARSIFY $(\mathcal{G}, \varepsilon, \alpha)$

---

**Input:** An $n$ vertex graph $\mathcal{G} = (V_\mathcal{G}, E_\mathcal{G})$, privacy parameters $(\alpha, \beta)$, approximation parameter $\varepsilon$.
**Output:** A Laplacian $L_{\widetilde{\mathcal{G}}}$.

1: **Privatize.** Construct a complete graph $\widehat{G}$ with weights $w_e$ defined as follows:

$$w_e := \begin{cases} +1 & \text{with probability } (1+\alpha)/2 & \text{if } e \in E_\mathcal{G} \\ -1 & \text{with probability } (1-\alpha)/2 & \text{if } e \in E_\mathcal{G} \\ +1 & \text{with probability } 1/2 & \text{if } e \notin E_\mathcal{G} \\ -1 & \text{with probability } 1/2 & \text{if } e \notin E_\mathcal{G} \end{cases}.$$

2: **Compute** $\bar{\mathcal{G}}$ using the linear program of Gupta et al. [23] on $\widehat{\mathcal{G}}$.
3: **Output** $\widetilde{\mathcal{G}}$ by running the algorithm of Benzur and Karger [8] on $\bar{\mathcal{G}}$.

---

The above theorem says that any cut can be approximated by our cut sparsifier up to $\widetilde{O}(\sqrt{n(n-s)s})$ error, while preserving $(\alpha, 0)$-differential privacy. This is an instance based bound and matches the accuracy achieved by our graph sparsification result (and the best possible [9, 54]) when $s = O(n)$.

# 6 Discussion

In this paper, we introduced private spectral sparsification of graphs. We gave efficient algorithms to compute one such sparsification. Our algorithm outputs a graph with $O(n/\varepsilon^2)$ edges, while preserving differential privacy. Our algorithmic framework allows us to compute an approximation to MAX-CUT and SPARSEST-CUT with better accuracy than previously known, and a first algorithm for learning differentially private Laplacian eigenmaps.

Our algorithm uses both importance sampling and random sketching. At a high level, sketching allows us to ensure privacy and importance sampling allows us to produce spectral sparsification. To the best of our knowledge, this is the first instance of using importance sampling in the context of privacy. Since important sampling is an important tool in non-private low-space algorithms [55], this we believe can lead to development of other private algorithms.

Our work differs from previous works that use random projections or graph sparsification [9, 54]. The only work that we are aware of which uses spectral sparsification in the context of differential privacy is that of Upadhyay [54] with an aim to improve the run-time efficiency of Blocki et al. [9]; however, their algorithm does not output a graph let alone a sparse graph. Hence, their approach cannot be used to approximate MAX-CUT or SPARSE-CUT – the only method to solve these problems would be to run all possible cut queries leading to an error of $O(n^{3/2}/\alpha)$. This follows because the error per cut query would be $O(\frac{\sqrt{n}|S|}{\varepsilon})$, and since there are $2^n$ possible cuts, an application of Chernoff bound results in worst case error $O(\frac{n^{3/2}}{\alpha})$, matching the guarantee of Gupta et al. [23].

On the other hand, Gupta et al. [23] give an algorithm to output a graph that preserves cut functions on a graph. However, their output does not preserve the spectral properties of the graph and so cannot be used in spectral applications of graphs, such as Laplacian eigenmap or learning Lipschitz functions on graphs. Moreover, their algorithm incurs large additive error. We also give a tighter analysis for a variant of their algorithm to achieve an instance based additive error.

**Acknowledgements.** This research was supported, in part, by NSF BIGDATA grants IIS-1546482 and IIS-1838139, and DARPA award W911NF1820267. This work was done when Jalaj Upadhyay was working as a postdoctoral researcher with Raman Arora at the Johns Hopkins University. Authors would like to thank Adam Smith, Lorenzo Orecchia, Cynthia Steinhardt, and Sarvagya Upadhyay for insightful discussions during the early stages of the project.

## Footnotes

[1]This is the standard neighboring relation used in Blocki et al. [9] (and references therein). The underlying reason for considering this privacy notion is as follows: since we do not restrict the weight on the graph, presence or absence of an edge, as required in standard edge-level privacy, can change the output drastically.

[2]Note that the number of non-zero singular values of $n$ vertex graph can be at most $n - 1$ [20], where $n$ is the number of nodes in the graphs. That is, if we consider node-level privacy, then the Laplacian of a graph $\mathcal{G}$ has at most $n$ singular values and that for neighboring graph $\mathcal{G}'$ is at most $n - 1$ singular values. Thus outputting any spectral sparsification would not preserve privacy.

[3]Deterministic algorithms cannot be differentially private. On the other hand, if not done carefully, the sampling probability that depends on the graph can itself leak privacy.

[4]Alon et al. [3] gave an algorithm for solving MAX-CUT for real-weight graphs using quadratic program (instead of semi-definite program based approach [21]), but that leads to an $O(\log n)$ multiplicative approximation.

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
