[Supplementary Material · supp.pdf]

# Supplementary Material to "On Differentially Private Graph Sparsification and Applications"

## A  Notation and Preliminaries

We let $\mathbb{N}$ to denote the set of natural numbers. We use capital letters to denote matrices and small letters to denote vectors. We denote by $\mathbf{0}^{m \times n}$ the all-zero $m \times n$ matrix and by $\mathbb{I}$ be the identity matrix. For a matrix $A$, we let $\mathsf{rk}(A)$ denote the rank, $\Delta(A)$ to denote its *determinant*, and $\mathsf{Tr}(A)$ denote the trace norm of the matrix $A$. The *singular-value decomposition* (SVD) of an $m \times n$ rank-$r$ matrix $A$ is a decomposition of $A$ as a product of three matrices, $A = U \Sigma V^\top$ such that $U \in \mathbb{R}^{m \times r}$ and $V \in \mathbb{R}^{n \times r}$ have orthonormal columns and $\Sigma \in \mathbb{R}^{r \times r}$ is a diagonal matrix with singular values of $A$ on its diagonal. We let $\mathbf{1}_S \in \{0,1\}^n$ denote the characteristic vector for a subset $S \subseteq [n]$. For a symmetric matrix, we write $A \succeq 0$ to denote a matrix whose singular values are greater than $0$ and $A \succeq B$ if $A - B \succeq 0$. The *Moore-Penrose pseudo-inverse* of a matrix $A = U \Sigma V^\top$ is denoted by $A^\dagger$ and has a SVD $A^\dagger = V \Sigma^\dagger U^\top$, where $\Sigma^\dagger$ consists of inverses of only non-zero singular values of $A$. We will use the following fact in our proof.

**Fact 8.** *For any two symmetric matrices $A$ and $B$, if $A \succeq B$, then $A^\dagger \preceq B^\dagger$.*

A matrix $A$ has a *left-inverse* (*right-inverse*, respectively) if and only if it has full column rank (full row rank, respectively). If $A$ has a left-inverse, then $A^\dagger = (A^\top A)^{-1} A^\top$ and if $A$ has right-inverse, then $A^\dagger = A^\top (A A^\top)^{-1}$.

We need the following results for the privacy proof.

**Theorem 9** (Lidskii Theorem). *Let $A, B$ be $n \times n$ Hermittian matrices. Then for any choice of indices $1 \le i_1 \le \cdots \le i_k \le n$,*

$$\sum_{j=1}^{k} \lambda_{i_j}(A + B) \le \sum_{j=1}^{k} \lambda_{i_j}(A) + \sum_{j=1}^{k} \lambda_{i_j}(B),$$

*where $\{\lambda_i(A)\}_{i=1}^{n}$ are the eigen-values of $A$ in decreasing order.*

### A.1  Gaussian Distribution

Given a random variable $x$, we denote by $\mathcal{N}(\mu, \rho^2)$ the fact that $x$ has a normal Gaussian distribution with mean $\mu$ and variance $\rho^2$. The Gaussian distribution is invariant under affine transformation, i.e., if $x \sim \mathcal{N}(\mu_1, \sigma_1)$ and $y \sim \mathcal{N}(\mu_2, \sigma_2)$, then $z = ax + by$ has the distribution $z \sim \mathcal{N}(a\mu_1 + b\mu_2, a\sigma_1^2 + b\sigma_2^2)$. By simple computation, one can verify that the tail of a standard Gaussian variable decays exponentially. More specifically, for a random variable $x \sim \mathcal{N}(0,1)$, we have $\Pr[|x| > t] \le 2e^{-t^2/2}$.

The multivariate Gaussian distribution is a generalization of univariate Gaussian distribution. Given an $m$-dimensional multivariate random variable, $X \sim \mathcal{N}(\mu, \Sigma)$ with mean $\mu \in \mathbb{R}^m$ and covariance matrix $\Sigma = \mathsf{E}[(X - \mu)(X - \mu)^T]$, the probability density function of a multivariate Gaussian is given by $\frac{1}{\sqrt{2\pi \bar{\Delta}(\Sigma)}} \exp\left(-\frac{1}{2} \mathbf{x}^T \Sigma^{-1} \mathbf{x}\right)$. It is easy to see from the description of the PDF that, in order to define the PDF corresponding to a multivariate Gaussian distribution, $\Sigma$ has to have full rank. If $\Sigma$ has a non-trivial kernel space, then the PDF is undefined. However, in this paper, we only need to compare the probability distribution of two random variables which are defined over the same subspace. Therefore, in those scenarios, we would restrict our attention to the subspace orthogonal to the kernel space of $\Sigma$.

Multivariate Gaussian distribution maintains many key properties of univariate Gaussian distribution. For example, any (non-empty) subset of multivariate normals is multivariate normal. Another key property that is important in our analysis is that linearly independent linear functions of multivariate normal random variables are multivariate normal random variables, i.e., if $Y = AX + b$, where $A$ is an $n \times n$ non-singular matrix and $b$ is a (column) $n$-vector of constants, then $Y \sim \mathcal{N}(A\mu + b, A\Sigma A^T)$.

## A.2 Graph Theory

We reserve the symbol $\mathcal{G}$, $\widehat{\mathcal{G}}$ and $\widetilde{\mathcal{G}}$ to denote a graph. We denote by $\mathcal{G}'$ the graph formed by adding an edge to the graph $\mathcal{G}$. For any $S \subseteq V(\mathcal{G})$, the *cut* of the set of vertices $S$, denoted it by $\Phi_{\mathcal{G}}(S)$, is the weight of the edges that are present between $S$ and $V \backslash S$. Since the vertex set is constant throughout this paper, we just write $V$ to denote $V(\mathcal{G})$.

**Definition 10** (Laplacian of graph). *For an undirected graph $\mathcal{G}$, its edge adjacency matrix $E_{\mathcal{G}} \in \mathbb{R}^{\binom{n}{2} \times n}$ is defined as follow: the row corresponding to edge $e = (u, v)$ with weight $w_e$ contains $w_e$ in column $u$, $(-w_e)$ in column $v$, and $0$ elsewhere. The Laplacian is then defined as $L_{\mathcal{G}} = E_{\mathcal{G}}^\top E_{\mathcal{G}}$.*

For any graph $\mathcal{G}$ with edge-adjacency matrix $E_{\mathcal{G}}$, consider its singular value decomposition $U \Sigma V^\top$. Then the effective resistance (also known as leverage score) of the edge $e_i$ is defined as $\tau_i := e_i^\top (E_{\mathcal{G}}^\top E_{\mathcal{G}})^\dagger e_i = e_i^\top L_{\mathcal{G}}^\dagger e_i$. It is well known that by sampling the edges (rows of $E_{\mathcal{G}}$) of $\mathcal{G}$ according to its leverage score, it is possible to obtain a graph $\widetilde{\mathcal{G}}$ such that $(1 - \varepsilon) L_{\widetilde{\mathcal{G}}} \preceq L_{\mathcal{G}} \preceq (1 + \varepsilon) L_{\widetilde{\mathcal{G}}}$. Our analysis critically uses the fact that sampling by leverage scores is robust against overestimate of leverage scores.

Many interesting properties of the Laplacian of a graph follows from this representation. For example, Laplacian of a graph is positive semi-definite, i.e., all the eigenvalues are non-negative. For a set $S$ of vertices, its cut-set is $\Phi_S(\mathcal{G}) = \mathbf{1}_S^T L_{\mathcal{G}} \mathbf{1}_S$.

We let $\lambda_i(\mathcal{G})$ denote the eigenvalues of $L_{\mathcal{G}}$ for $1 \le i \le n$. Next we present few lemmata that are useful in our analysis. We analyze multivariate Gaussian distributions that are linear combination of the Laplacian of two graphs. In order to analyze the two distributions, the corresponding covariance matrices must span the same subspace. The first lemma allows us to work on the same subspace, that is, the subspace orthogonal to $\mathsf{Span}\{\mathbf{1}\}$.

**Lemma 11.** *Let $0 = \lambda_1(\mathcal{G}) \le \lambda_2(\mathcal{G}) \cdots \le \lambda_n(\mathcal{G})$ be the $n$ eigenvalues of $L_{\mathcal{G}}$. Then $\mathcal{G}$ is connected iff $\lambda_2 > 0$ and the kernel space of a connected graph is $\mathsf{Span}\{\mathbf{1}\}$. More generally, if a graph has $k$ components, then the multiplicity of the eigenvalue $0$ is $k$.*

The following two lemmata are useful in giving the upper bound while proving the $\mathsf{DP}$ of our sanitizer.

**Lemma 12.** *Let $\mathcal{G}$ and $\mathcal{G}'$ be two graphs, where $\mathcal{G}'$ is obtained from $\mathcal{G}$ by adding one edge joining two distinct vertices of $\mathcal{G}$. Then*

$$\lambda_2(\mathcal{G}) \le \lambda_2(\mathcal{G}') \le \lambda_2(\mathcal{G}) + 2.$$

**Lemma 13.** *Let $\mathcal{G}'$ be formed by adding an edge $(u, v)$ to $\mathcal{G}$. For any vector $\mathbf{x} \in \mathbb{R}^n$, we have $\mathsf{Tr}(L_{\mathcal{G}'}) \le \mathsf{Tr}(L_{\mathcal{G}}) + 2$.*

The following lemma is particularly useful in arguing that the lowest non-zero eigenvalues of all the graphs is bounded from below by a constant (which is the second smallest eigenvalue of an expander).

**Lemma 14** (Eigenvalue Interlacing). *Let $\mathcal{G}$ and $\mathcal{G}'$ be two graphs, where $\mathcal{G}'$ is obtained from $\mathcal{G}$ by adding one edge joining two distinct vertices of $\mathcal{G}$. Then*

$$\lambda_i(\mathcal{G}) \le \lambda_i(\mathcal{G}') \le \lambda_{i+1}(\mathcal{G}).$$

*In particular, if $\mathcal{H}$ be a subgraph of $\mathcal{G}$, then $\lambda_i(\mathcal{H}) \le \lambda_i(\mathcal{G}) \forall 1 \le i \le n$.*

## A.3 Differential Privacy

It is important to make a note that, unlike the traditional notions of privacy in cryptography, differential privacy is a measure of privacy loss. This allows one to quantify and compare various techniques and answer questions like, which is the most accurate mechanism for a given privacy loss, or which mechanism provides most privacy for a given error in the accuracy.

There are two settings in which a privacy preserving mechanism can be designed: non-interactive and interactive setting. In the *non-interactive* setting the curator publishes some sanitized database, and the database is not used further. The precise answers released by the curator might be affected by the sanitized database. As the database is never used again, the curator may as well destroy the database once it has published the sanitized form.

On the other hand, in the *interactive* setting, the curator sits between the analyst and the database, and queries are posed by the analyst in an adaptive manner. The curator must respond to these queries in a fashion so as to protect the privacy of database.

In this paper, all our algorithms is *non-interactive*. Differential privacy satisfies many required properties of any privacy guarantee. For example, differential privacy is immune to auxiliary information whether the auxiliary information is current or any side information an adversary can access in the future. We call this property the *robustness* of differential privacy. Another key property that differentially private mechanisms "composes" really well. This allows us to construct complicated mechanism from basic and simpler private mechanisms. We next cover both these properties in more details.

**Robustness of Differential Privacy.** One of the key features of differential privacy is that it is preserved under arbitrary post-processing, i.e., an analyst, without additional information about the private database, cannot compute a function that makes an output less differentially private. In other words,

**Lemma 15** (Post-processing [17])**.** *Let $\mathfrak{M}(\mathbf{D})$ be an $(\alpha, \beta)$-differential private mechanism for a database $\mathbf{D}$, and let $h$ be any function, then any mechanism $\mathfrak{M}' := h(\mathfrak{M}(\mathbf{D}))$ is also $(\alpha, \beta)$-differentially private for the same set of tasks.*

*Proof.* Let $\mathfrak{M}$ be a differentially private mechanism. Let $\mathsf{Range}(\mathfrak{M})$ denote the range the of $\mathfrak{M}$.Let $R$ be the range of the function $h(\cdot)$. Without loss of generality, we assume that $h(\cdot) : \mathsf{Range}(\mathfrak{M}) \to \mathcal{R}$ is a deterministic function. This is because any randomized function can be decomposed into a convex combination of deterministic function, and a convex combination of differentially private mechanisms is differentially private. Fix any pair of neighbouring data-sets $\mathbf{DB}$ and $\widetilde{\mathbf{DB}}$ and an event $S \subseteq \mathcal{R}$. Let $T = \{y \in \mathsf{Range}(\mathfrak{M}) : f(r) \in S\}$. Then

$$\Pr[f(\mathfrak{M}(\mathbf{DB})) \in S] = \Pr[\mathfrak{M}(\mathbf{DB}) \in T]$$
$$\leq \exp(\alpha)\Pr[\mathfrak{M}(\widetilde{\mathbf{DB}}) \in T] + \beta$$
$$= \exp(\alpha)\Pr[f(\mathfrak{M}(\widetilde{\mathbf{DB}})) \in S] + \beta$$

as required. $\qquad\qquad\square$

**Composition.** Before we begin, we discuss what does it mean by the term "composition" of differentially private mechanism. The composition that we consider covers the following two cases:

1. Repeated use of differentially private mechanism on the same database.
2. Repeated use of differentially private mechanism on different database that might contain information relating to a particular individual.

The first case covers the case when we wish to use the same mechanism multiple times while the second case covers the case of cumulative loss of privacy of a single individual whose data might be spread across many databases.

It is easy to see that the composition of pure differentially private mechanisms yields another pure differentially private mechanism, i.e., composition of an $(\alpha_1, 0)$-differentially private and an $(\alpha_2, 0)$-differentially private mechanism results in an $(\alpha_1 + \alpha_2, 0)$-differentially private mechanism. In other words, the privacy guarantee depreciates linearly with the number of compositions. In the case of approximate differential privacy, we can improve on the degradation of $\alpha$ parameter at the cost of slight depreciation of the $\beta$ factor. We use this strengthening in our proofs. In our proofs of differential privacy, we prove that each row of the published matrix preserves $(\alpha_0, \beta_0)$-differential privacy for some appropriate $\alpha_0, \beta_0$, and then invoke a composition theorem by [17] to prove that the published matrix preserves $(\alpha, \beta)$-differential privacy. The following theorem is the composition theorem that we use.

**Theorem 16** (Composition theorem [17])**.** *Let $\alpha_0, \beta_0 \in (0,1)$, and $\beta' > 0$. If $\mathfrak{M}_1, \cdots, \mathfrak{M}_\ell$ are each $(\alpha, \beta)$-differential private mechanism, then the mechanism $\mathfrak{M}(\mathbf{D}) := (\mathfrak{M}_1(\mathbf{D}), \cdots, \mathfrak{M}_\ell(\mathbf{D}))$ releasing the concatenation of each algorithm is $(\alpha', \ell\beta + 0 + \beta')$-differentially private for $\alpha' < \sqrt{2\ell \ln(1/\beta')}\alpha_0 + 2\ell\alpha_0^2$.*

A proof of this theorem could be found in [17, Chapter 3].

# B Missing Proofs of Section 3 and Section 5

Before we give a detail proof of our theorem, we make few important remarks regarding the result. Our bound (Theorem 3) states that the Laplacian of the graph is perturbed by an additive term, $\frac{w}{n}L_n$, due to privacy considerations. Given the choice of $w$, the scale of this additive perturbation is $\widetilde{O}(1/\sqrt{n})$. To better appreciate the bound, let us consider the following simple procedure where we obfuscate the edges through randomization and then run a non-private sparsification algorithm. This procedure would incur an approximation of the form $(1-\varepsilon)(L_n + L_{\mathcal{G}}) \preceq L_{\widetilde{\mathcal{G}}} \preceq (1+\varepsilon)(L_n + L_{\mathcal{G}})$, i.e., the scale of the additive perturbation matrix (which is still $L_n$) is $O(1)$. In Appendix E, we show that a solution based on the exponential mechanism will suffer from a similar problem. In light of this, our result yields a significant improvement over those alternatives. Finally, we note that returning a complete graph is not an option because the spectrum of $(1-\varepsilon)L_n$ might not be a lower bound on the spectrum of $\mathcal{G}$.

Theorem 3 does not put any restriction on $\mathcal{G}$. In particular, $\mathcal{G}$ can be a weighted graph with possibly very large spectrum compared to $L_n/\sqrt{n}$. Further, larger the weight, larger would be the value of MAX-CUT or $\Phi_{\mathcal{G}}(S)$. In fact, many naturally occurring graphs have large weights. One such graph arises in the discovery of metabolic pathways in genomic networks [26], where weights indicate strength between genes [15].

## B.1 Utility Proof

We now prove the utility proof. We first prove that $\widetilde{\mathcal{G}}'$ is a graph that spectrally approximates $\bar{\mathcal{G}}$. The complication in the proof arise from the fact that $D$ is not formed by the effective resistances of $\bar{\mathcal{G}}$, but rather the approximation of effective resistances of $\widehat{\mathcal{G}}$.

We construct matrix random variables corresponding to the graph $\bar{\mathcal{G}}$ defined in Algorithm 1. Let $e_i$ be the edges in $\bar{\mathcal{G}}$ defined in Algorithm 1. Recall that $p_i = c\widetilde{\tau}_i\varepsilon^{-2}\log(n/\delta)$. We define the following random variables corresponding to each edge,

$$X_i := \begin{cases} \frac{e_i e_i^\top}{p_i} & \text{with probability } p_i \\ 0 & \text{with probability } 1 - p_i \end{cases}.$$

Then $Y = \sum_i X_i = E_{\bar{\mathcal{G}}}^\top D E_{\bar{\mathcal{G}}}$. It is easy to see that $\mathsf{E}[Y] = L_{\bar{\mathcal{G}}}$, where expectation is over $X_i$.

Then, we can finish the proof by appealing to the matrix Bernstein inequality if we can bound $X_i \preceq R \cdot \mathsf{E}[Y]$. When $p_i = 1$, $X_i = e_i e_i^\top$ with probability 1, which is the same as selecting and summing $c\varepsilon^{-2}\log(n/\delta)$ random variables each equal to $\frac{e_i e_i^\top}{c\varepsilon^{-2}\log(n/\delta)}$, and hence $X_i \preceq \frac{L_{\bar{\mathcal{G}}}}{c\varepsilon^{-2}\log(n/\delta)}$. When $p_i < 1$, we can bound $X_i$ as follows.

**Proposition 17.** *Let $X_i, \bar{\mathcal{G}}$ be as defined above. Then with probability $1-\delta$, $X_i \preceq O\left(\frac{\varepsilon^2}{c\log(n/\delta)}\right)L_{\bar{\mathcal{G}}}$.*

*Proof.* Since $\bar{\mathcal{G}}$ is a connected graph, the all ones vector, denoted $\mathbf{1}$, spans the kernel of the column space of $L_{\bar{\mathcal{G}}}$ [20]. Then, for an $x \in \mathbb{R}^n$, write $x = x_\| + x_\perp$, where $x_\|$ is in the span of column space of $L_{\widehat{\mathcal{G}}}$ and $x_\perp$ is in the span of $\mathbf{1}$. In other words, $\langle x_\|, \mathbf{1}\rangle = 0$. Let $U\Sigma^2 U^\top$ be the singular value decomposition of $L_{\bar{\mathcal{G}}}$. So,

$$\begin{aligned} x^\top L_{\bar{\mathcal{G}}} x &= (x_\| + x_\perp)^\top L_{\bar{\mathcal{G}}}(x_\| + x_\perp) = x_\|^\top L_{\bar{\mathcal{G}}} x_\| \\ &= y^\top U\Sigma^{-1}U^\top L_{\bar{\mathcal{G}}}U\Sigma^{-1}U^\top y \\ &= y^\top U\Sigma^{-1}U^\top U\Sigma^2 U^\top U\Sigma^{-1}U^\top y = \|y\|_2^2 \end{aligned}$$

for some $y$ in the column space of $L_{\bar{\mathcal{G}}}$. Now with probability at least $1 - \delta$,

$$
\begin{aligned}
x^\top e_i e_i^\top x &= (x_\| + x_\perp)^\top e_i e_i^\top (x_\| + x_\perp) \\
&= x_\|^\perp e_i e_i^\top x_\| \\
&= y^\top U \Sigma^{-1} U^\top e_i e_i^\top U \Sigma^{-1} U^\top y \\
&\leq \mathsf{Tr}(U \Sigma^{-1} U^\top e_i e_i^\top U \Sigma^{-1} U^\top) \| y \|_2^2 \\
&= e_i^\top L_{\bar{\mathcal{G}}}^\dagger e_i \| y \|_2^2 \\
&\leq C e_i^\top L_{\widehat{\mathcal{G}}}^\dagger e_i \| y \|_2^2 \\
&= C \widetilde{\tau}_i x^\top L_{\bar{\mathcal{G}}} x
\end{aligned}
$$

for $C > 2$. Here the inequality follows from the subadditivity of trace norm, Fact 8, equation (2), and the final equality follows from the definition of $\widetilde{\tau}_i$. Since $p_i = c \widetilde{\tau}_i \varepsilon^{-2} \log(n/\delta)$, we have

$$
X_i \preceq \frac{e_i e_i^\top}{p_i} \preceq \frac{e_i e_i^\top}{c \widetilde{\tau}_i \varepsilon^{-2} \log(n/\delta)} \preceq \frac{L_{\bar{\mathcal{G}}}}{c \varepsilon^{-2} \log(n/\delta)}
$$

as required. $\qquad\qquad\square$

Using matrix Bernstein inequality [53], we have

$$
\Pr\left[(1-\varepsilon)Y \preceq L_{\bar{\mathcal{G}}} \preceq (1+\varepsilon)Y\right] \geq 1 - n e^{-c\varepsilon^{-2}\log(n/\delta)/3}
$$

for large enough $c$. Now $E_{\bar{\mathcal{G}}}^\top D E_{\bar{\mathcal{G}}} = Y$ implies

$$
(1-\varepsilon) E_{\bar{\mathcal{G}}}^\top D E_{\bar{\mathcal{G}}} \preceq L_{\bar{\mathcal{G}}} \preceq (1+\varepsilon) E_{\bar{\mathcal{G}}}^\top D E_{\bar{\mathcal{G}}}.
$$

with probability $1 - \delta$. That is, $E_{\bar{\mathcal{G}}}^\top D E_{\bar{\mathcal{G}}}$ is a private spectral approximation of $\bar{\mathcal{G}}$ with Laplacian $\left(\frac{w}{n} L_n + \left(1 - \frac{w}{n}\right) L_{\mathcal{G}}\right)$. $E_{\bar{\mathcal{G}}}^\top D E_{\bar{\mathcal{G}}}$ can be the Laplacian of a dense graph. Using the result of [35], we have $(1-\varepsilon) L_{\widetilde{\mathcal{G}}} \preceq E_{\bar{\mathcal{G}}}^\top D E_{\bar{\mathcal{G}}} \preceq (1+\varepsilon) L_{\widetilde{\mathcal{G}}}$ and that $L_{\widetilde{\mathcal{G}}}$ has $O(n/\varepsilon^2)$ edges. In total, this implies that

$$
\Pr\left[(1-\varepsilon)^2 L_{\widetilde{\mathcal{G}}} \preceq L_{\bar{\mathcal{G}}} \preceq (1+\varepsilon)^2 L_{\widetilde{\mathcal{G}}}\right] \geq 1 - 2\delta.
$$

Since the solution of SDP-1 is $\lambda = c\sqrt{\frac{\log n \log(1/\delta)}{n\alpha^2}}$, we have

$$
L_{\widetilde{\mathcal{G}}'} - c\sqrt{\frac{\log n \log(1/\delta)}{n\alpha^2}} L_n \preceq L_{\bar{\mathcal{G}}} \preceq L_{\widetilde{\mathcal{G}}'} + c\sqrt{\frac{\log n \log(1/\delta)}{n\alpha^2}} L_n.
$$

Another application of matrix Bernstein inequality gives us

$$
\Pr\left[L_{\widehat{\mathcal{G}}} - \rho\sqrt{\frac{\log n \log(1/\delta)}{n\alpha^2}} L_n \preceq L_{\widetilde{\mathcal{G}}'} \preceq L_{\widehat{\mathcal{G}}} + \rho\sqrt{\frac{\log n \log(1/\delta)}{n\alpha^2}} L_n\right] \geq 1 - \delta. \qquad (2)
$$

for some small constant $\rho > 0$. Now by construction,

$$
L_{\widehat{\mathcal{G}}} := \frac{w}{n} L_n + \left(1 - \frac{w}{n}\right) L_{\mathcal{G}}
$$

for $w = O(\sqrt{\frac{\log n \log(1/\delta)}{n\alpha^2}})$. Combining all these partial orderings gives us the accuracy bound.

### B.2 Privacy Proof of Algorithm 1

In this section, we give a detail proof of the privacy guarantee of Algorithm 1. First note that computing $L_{\widetilde{\mathcal{G}}'}$ and $R$ is $(\alpha/3, \beta/3)$-differentially private due to Gaussian mechanism [17] and Johnson-Lindenstrauss mechanism [9].

**Lemma 18.** *Let $M \in \mathbb{R}^{(m+n)\times t}$ be a random Gaussian matrix with every entries sampled i.i.d. from $\mathcal{N}(0, 1/t)$. Let $E_{\widehat{\mathcal{G}}}$ be a graph formed by overlaying the input graph with a complete graph with weights $w/n$, where $w = \frac{\sqrt{t\log(1/\beta)}}{\alpha}\log(1/\beta)$. Then*

$$
B := \begin{pmatrix} E_{\widehat{\mathcal{G}}} & w\mathbb{I} \end{pmatrix} M
$$

*is $(\alpha, \beta)$-differentially private.*

*Proof.* We spot a subtle mistake in the previous analysis of privacy by [9, 54]. As in their proof, we also prove that each row of the published matrix preserves $(\alpha_0, \beta_0)$-differential privacy for some appropriate $\alpha_0, \beta_0$, and then invoke Theorem 16 to prove that the published matrix preserves $(\alpha, \beta)$-differential privacy. Denote by $\widehat{A} = \begin{pmatrix} E_{\widehat{\mathcal{G}}} & w\mathbb{I}_m \end{pmatrix}$ and by $\widehat{A}' = \begin{pmatrix} E'_{\widehat{\mathcal{G}}} & w\mathbb{I}_m \end{pmatrix}$, where $\widehat{\mathcal{G}}$ and $\widehat{\mathcal{G}}'$ differ in one edge of weight 1. Note that both $\widehat{A}$ and $\widehat{A}'$ are full rank matrix because of the construction. This implies that the affine transformation of the multi-variate Gaussian is well-defined (both the covariance $(\widehat{A}\widehat{A}^\top)^{-1}$ has a full rank and the $\Delta(\widehat{A}\widehat{A}^\top)$ is non-zero). That is, the PDF of the distributions of the columns, corresponding to $\widehat{A}$ and $\widehat{A}'$, is just a linear transformation of $\mathcal{N}(\mathbf{0}, \mathbb{I})$. Let $y \sim \mathcal{N}(\mathbf{0}, \mathbb{I})$ be a column vector. Let $x$ is sampled either from $\widehat{A}y$ or $\widehat{A}'y$. Then the corresponding PDFs are as follows:

$$\mathsf{PDF}_{\widehat{A}y}(x) = \frac{e^{(-\frac{1}{2}x(\widehat{A}\widehat{A}^\top)^{-1}x^\top)}}{\sqrt{(2\pi)^d \Delta(\widehat{A}\widehat{A}^\top)}}, \quad \mathsf{PDF}_{\widehat{A}'y}(x) = \frac{e^{(-\frac{1}{2}x(\widehat{A}'\widehat{A}'^\top)^{-1}x^\top)}}{\sqrt{(2\pi)^d \Delta(\widehat{A}'\widehat{A}'^\top)}}$$

Let

$$\alpha_0 = \frac{\alpha}{\sqrt{4t \ln(1/\beta) \log(1/\beta)}}, \quad \beta_0 = \frac{\beta}{2t},$$

where $t = n$ in the case of PRIVATE-SPARSIFY. We prove that every row of the published matrix is $(\alpha_0, \beta_0)$-differentially private; the theorem follows from Theorem 16. Let $x$ be sampled either from $\mathcal{N}(0, \widehat{A}\widehat{A}^\top)$ or $\mathcal{N}(0, \widehat{A}'\widehat{A}'^\top)$. It is straightforward to see that the combination of Claim 19 and Claim 20 below proves differential privacy for a row of published matrix. The lemma then follows by an application of Theorem 16 and our choice of $\alpha_0$ and $\beta_0$. We prove a stronger guarantee than what is required, i.e., we prove privacy when $\widehat{A}' - \widehat{A} = uv^\top$ for unit vectors $u, v$. Privacy for various theorems in this paper can be realized by setting $u$ to be a standard basis vector.

**Claim 19.** *Let $\widehat{A}$, $\widehat{A}'$ and $\alpha_0$ be as defined above. Then*

$$e^{-\alpha_0} \le \sqrt{\frac{\Delta(\widehat{A}\widehat{A}^\top)}{\Delta(\widehat{A}'\widehat{A}'^\top)}} \le e^{\alpha_0}.$$

*Proof.* The claim follows simply as in [9]. More concretely, we have $\Delta(\widehat{A}\widehat{A}^\top) = \prod_i \sigma_i^2$, where $\sigma_1 \ge \cdots \ge \sigma_m \ge \sigma_{\min}(\widehat{A})$ are the singular values of $\widehat{A}$. Let $\widetilde{\sigma}_1 \ge \cdots \ge \widetilde{\sigma}_m \ge \sigma_{\min}(\widehat{A}')$ be its singular value for $\widehat{A}'$. The matrix $E = \widehat{A}' - \widehat{A}$ has only one singular value.

Since the singular values of $\widehat{A} - \widehat{A}'$ and $\widehat{A}' - \widehat{A}$ are the same, Lidskii's theorem (Theorem 9) gives $\sum_i (\sigma_i - \widetilde{\sigma}_i) \le 1$. Therefore, with probability $1 - \beta_0$,

$$\sqrt{\prod_{i:\widetilde{\sigma}_i \ge \sigma_i} \frac{\widetilde{\sigma}_i^2}{\sigma_i^2}} = \prod_{i:\widetilde{\sigma}_i \ge \sigma_i} \left(1 + \frac{\widetilde{\sigma}_i - \sigma_i}{\sigma_i}\right) \le \exp\left(\frac{\varepsilon}{32\sqrt{t \log(2/\beta)} \log(t/\beta)} \sum_{i:\widetilde{\sigma}_i \ge \sigma_i} (\widetilde{\sigma}_i - \sigma_i)\right) \le e^{\alpha_0/2},$$

where the first inequality follows from the fact that $1 + x \le e^x$ for $x \ge 0$ and the fact that $\sigma_i \ge \sigma_{\min}$.

The case for all $i \in [m]$ when $\widetilde{\sigma}_i \le \sigma_i$ follows similarly as the singular values of $E$ and $-E$ are the same. This completes the proof of Claim 19. $\qquad\square$

**Claim 20.** *Let $\widehat{A}, \alpha_0$, and $\beta_0$ be as defined earlier. Let $y \sim \mathcal{N}(\mathbf{0}, \mathbb{I})$. If $x$ is sampled either from $\widehat{A}y$ or $\widehat{A}'y$, then we have*

$$\Pr\left[\left|x^\top (\widehat{A}\widehat{A}^\top)^{-1} x - x^\top (\widehat{A}'\widehat{A}'^\top)^{-1} x\right| \le \alpha_0\right] \ge 1 - \beta_0.$$

*Proof.* Without any loss of generality, we can assume $x = \widehat{A}y$. The case for $x = \widehat{A}'y$ is analogous. Let $\widehat{A}' - \widehat{A} = uv^\top$. Then

$$x^\top (\widehat{A}\widehat{A}^\top)^{-1} x - x^\top (\widehat{A}'\widehat{A}'^\top)^{-1} x = x^\top (\widehat{A}\widehat{A}^\top)^{-1} (\widehat{A}'\widehat{A}'^\top)(\widehat{A}'\widehat{A}'^\top)^{-1} x - x^\top (\widehat{A}'\widehat{A}'^\top)^{-1} x$$

$$= x^\top (\widehat{A}\widehat{A}^\top)^{-1} (\widehat{A} + uv^\top)(\widehat{A}^\top + vu^\top)(\widehat{A}'\widehat{A}')^{-1} x - x^\top (\widehat{A}'\widehat{A}')^{-1} x$$

$$= x^\top \left[(\widehat{A}\widehat{A}^\top)^{-1}(\widehat{A}uv^\top + vu^\top \widehat{A}^\top + uu^\top)(\widehat{A}'\widehat{A}'^\top)^{-1}\right] x.$$

Previous analysis failed to bound the last term due to $uu^\top$. Using the singular value decomposition of $\widehat{A} = U_A \Sigma_A V_A^\top$ and $\widehat{A}' = \widetilde{U}_A \widetilde{\Sigma}_A \widetilde{V}_A^\top$, we have

$$\left(x^\top (U_A \Sigma_A^{-1} V_A^\top) u\right) \left(v^\top (\widetilde{U}_A \widetilde{\Sigma}_A^{-2} \widetilde{U}_A^\top) x\right) + \left(x^\top (U_A \Sigma_A^{-2} U_A^\top) v\right) \left(u^\top (\widetilde{V}_A \widetilde{\Sigma}_A^{-1} \widetilde{U}_A^\top) x\right)$$

$$+ \left(x^\top (U_A \Sigma_A^{-2} U_A^\top) u\right) \left(u^\top (\widetilde{U}_A \widetilde{\Sigma}_A^{-2} \widetilde{U}_A^\top) x\right)$$

Since $x \sim \widehat{A} y$, where $y \sim \mathcal{N}(\mathbf{0}, \mathbb{I})$, we can write the above expression as $\tau_1 \tau_2 + \tau_3 \tau_4$, where

$$\tau_1 = \left(y^\top \widehat{A}^\top (U_A \Sigma_A^{-1} V_A^\top) u\right), \quad \tau_2 = \left(v^\top (\widetilde{U}_A \widetilde{\Sigma}_A^{-2} \widetilde{U}_A^\top) \widehat{A} y\right), \quad \tau_3 = \left(y^\top \widehat{A}^\top (U_A \Sigma_A^{-2} U_A^\top) v\right)$$

$$\tau_4 = \left(u^\top (\widetilde{V}_A \widetilde{\Sigma}_A^{-1} \widetilde{U}_A^\top) \widehat{A} y\right), \quad \tau_5 = \left(y^\top \widehat{A}^\top (U_A \Sigma_A^{-2} U_A^\top) u\right), \quad \tau_6 = \left(u^\top (\widetilde{U}_A \widetilde{\Sigma}_A^{-2} \widetilde{U}_A^\top) \widehat{A} y\right).$$

From the construction, we have $\|\widetilde{\Sigma}_A\|_2, \|\Sigma_A\|_2 \geq w$. Using the SVD of $\widehat{A}$ and $\widehat{A} - \widehat{A}' = vu^\top$, and that every term $\tau_i$ in the above expression is a linear combination of a Gaussian, i.e., each term is distributed as per $\mathcal{N}(0, \|\tau_i\|^2)$, we calculate $\|\tau_i\|$ as below.

$$\|\tau_1\|_2 = \|(V_A \Sigma_A U_A^\top)(U_A \Sigma_A^{-1} V_A^\top) u\|_2 \leq \|u\|_2 \leq 1,$$

$$\|\tau_2\|_2 = \|v^\top (\widetilde{U}_A \widetilde{\Sigma}_A^{-2} \widetilde{U}_A^\top)(\widetilde{U}_A \widetilde{\Sigma}_A \widetilde{V}_A^\top - vu^\top)\|_2$$

$$\leq \|v^\top (\widetilde{U}_A \widetilde{\Sigma}_A^{-2} \widetilde{U}_A^\top) \widetilde{U}_A \widetilde{\Sigma}_A \widetilde{U}_A^\top\|_2 + \|v^\top (\widetilde{U}_A \widetilde{\Sigma}_A^{-2} \widetilde{U}_A^\top) vu^\top\|_2 \leq \frac{1}{w} + \frac{1}{w^2},$$

$$\|\tau_3\|_2 = \|(V_A \Sigma_A U_A^\top)(U_A \Sigma_A^{-2} U_A^\top) v\|_2 \leq \|\Sigma_A^{-1}\|_2 \leq \frac{1}{w},$$

$$\|\tau_4\|_2 = \|u^\top (\widetilde{V}_A \widetilde{\Sigma}_A^{-1} \widetilde{U}_A^\top)(\widetilde{U}_A \widetilde{\Sigma}_A \widetilde{V}_A^\top - vu^\top)\|_2$$

$$\leq \|u^\top (\widetilde{V}_A \widetilde{\Sigma}_A^{-1} \widetilde{U}_A^\top)(\widetilde{U}_A \widetilde{\Sigma}_A \widetilde{V}_A^\top\|_2 + \|u^\top (\widetilde{V}_A \widetilde{\Sigma}_A^{-1} \widetilde{U}_A^\top) v\|_2 \leq 1 + \frac{1}{w}$$

$$\|\tau_5\|_2 = \|(V_A \Sigma_A U_A^\top)(U_A \Sigma_A^{-2} U_A^\top) u\|_2 \leq \|\Sigma_A^{-1}\|_2 \leq \frac{1}{w},$$

$$\|\tau_6\|_2 = \|u^\top (\widetilde{U}_A \widetilde{\Sigma}_A^{-2} \widetilde{U}_A^\top)(\widetilde{U}_A \widetilde{\Sigma}_A \widetilde{V}_A^\top - vu^\top)\|_2$$

$$\leq \|u^\top (\widetilde{U}_A \widetilde{\Sigma}_A^{-2} \widetilde{U}_A^\top) \widetilde{U}_A \widetilde{\Sigma}_A \widetilde{U}_A^\top\|_2 + \|u^\top (\widetilde{U}_A \widetilde{\Sigma}_A^{-2} \widetilde{U}_A^\top) vu^\top\|_2 \leq \frac{1}{w} + \frac{1}{w^2}.$$

Using the concentration bound on the Gaussian distribution, each term, $\tau_1, \tau_2, \tau_3, \tau_4, \tau_5$, and $\tau_6$, is less than $\|\tau_i\| \ln(4/\beta_0)$ with probability $1 - \beta_0/2$. The second claim follows from the following inequality:

$$\mathsf{Pr}\left[\left|x^\top (\widehat{A}\widehat{A}^\top)^{-1} x - x^\top (\widehat{A}'^\top \widehat{A}')^{-1} x\right| \leq 2 \left(\frac{1}{w} + \frac{1}{w^2} + \frac{1}{w^3}\right) \ln(4/\beta_0) \leq \alpha_0\right] \geq 1 - \beta_0,$$

where the second inequality follows from the choice of $w$. $\qquad\square$

Combining the two claims proves the lemma. $\qquad\square$

### B.3 Finishing the Proof of Theorem 3

In Lemma 21, we prove the algorithm is $(\alpha, \beta)$-differentially private. We prove that the effective resistance computed in PRIVATE-SPARSIFY is an overestimate in Lemma 22.

**Lemma 21.** $L_{\widetilde{\mathcal{G}}}$ *outputted by* PRIVATE-SPARSIFY *is* $(\alpha, \beta)$-*differentially private with respect to the edge-level privacy.*

*Proof.* We first prove the privacy guarantee. There are two times when the graph is used in the algorithm PRIVATE-SPARSIFY in Step 3. We produce a sketch of the form $B = AM$, where $M$ is a random Gaussian matrix and $A = \begin{pmatrix} E_{\widehat{\mathcal{G}}} & w\mathbb{I} \end{pmatrix}$. $B$ is $(\alpha/3, \beta/3)$-differentially private for the choice of $w$ due to Lemma 18. The second place where we use the graph is $NE_{\widehat{\mathcal{G}}}$. Again, here by the choice of $w$, we have $(\alpha/3, \beta/3)$-differentially private due to [9, 54]. Finally, we use the graph to compute $\widetilde{\mathcal{G}}'$. This is private due to Gaussian mechanism [17]. Using the composition theorem [18] and post-processing property (Lemma 15) of differential privacy, the result follows. $\qquad\square$

We next prove that $\widetilde{\tau}_i$ is an overestimate of the original $\tau_i$ corresponding to the edges in $\widehat{\mathcal{G}}$.

**Lemma 22.** *Let $\tau_i$ be the true leverage score of the $i$-th row of $E_{\bar{\mathcal{G}}}$. Then with probability $1 - \delta$, $\widetilde{\tau}_i \geq \tau_i$, where $\widetilde{\tau}_i$ is as computed in PRIVATE-SPARSIFY. Furthermore, $\sum_{i=1}^{\binom{n}{2}} \widetilde{\tau}_i = O(n^2)$.*

*Proof.* We first prove the second part of the lemma. Let $M = \begin{pmatrix} M_1 \\ M_2 \end{pmatrix}$, where $M_1 \in \mathbb{R}^{n \times n}$ and $M_2 \in \mathbb{R}^{m \times n}$. Then

$$
\begin{aligned}
\sum_i \widetilde{\tau}_i &= \mathsf{Tr}\left( \begin{pmatrix} E_{\widehat{\mathcal{G}}} & w\mathbb{I} \end{pmatrix} \begin{pmatrix} M_1 \\ M_2 \end{pmatrix} L_G^\dagger \begin{pmatrix} M_1^\top & M_2^\top \end{pmatrix} \begin{pmatrix} E_{\widehat{\mathcal{G}}}^\top \\ w\mathbb{I} \end{pmatrix} \right) \\
&\leq \mathsf{Tr}\left( \begin{pmatrix} L_{\widehat{\mathcal{G}}} & 0 \\ 0 & w^2\mathbb{I} \end{pmatrix} \begin{pmatrix} M_1 \\ M_2 \end{pmatrix} L_{\widehat{\mathcal{G}}}^\dagger \begin{pmatrix} M_1^\top & M_2^\top \end{pmatrix} \right) \\
&= \mathsf{Tr}\left( \begin{pmatrix} L_{\widehat{\mathcal{G}}} & 0 \\ 0 & w^2\mathbb{I} \end{pmatrix} \begin{pmatrix} M_1 L_{\widehat{\mathcal{G}}}^\dagger M_1^\top & 0 \\ 0 & M_2 L_G^\dagger M_2^\top \end{pmatrix} \right) \\
&= \mathsf{Tr}\left( \begin{pmatrix} L_{\widehat{\mathcal{G}}} M_1 L_{\widehat{\mathcal{G}}}^\dagger M_1^\top & 0 \\ 0 & w^2 M_2 L_G^\dagger M_2^\top \end{pmatrix} \right) \\
&= \mathsf{Tr}\left( M_1^\top L_{\widehat{\mathcal{G}}} M_1 L_{\widehat{\mathcal{G}}}^\dagger \right) + w^2 \mathsf{Tr}\left( M_2^\top M_2 L_{\widehat{\mathcal{G}}}^\dagger \right) \\
&= O(n^2).
\end{aligned}
$$

The last equality holds because $\mathsf{Tr}(M_1^\top L_{\widehat{\mathcal{G}}} M_1 L_{\widehat{\mathcal{G}}}^\dagger)$ can be written as $\mathsf{Tr}(U_{\widehat{\mathcal{G}}} M_1^\top U_{\widehat{\mathcal{G}}}^\top \Sigma_{\widehat{\mathcal{G}}} U_{\widehat{\mathcal{G}}} M_1 U_{\widehat{\mathcal{G}}}^\top \Sigma_{\widehat{\mathcal{G}}}^\dagger)$, where $L_{\widehat{\mathcal{G}}} = U_{\widehat{\mathcal{G}}}^\top \Sigma_{\widehat{\mathcal{G}}} U_{\widehat{\mathcal{G}}}$. Using the symmetric property of Gaussian distribution, this is identically distributed as $\mathsf{Tr}(M_1^\top \Sigma_{\widehat{\mathcal{G}}} M_1 \Sigma_{\widehat{\mathcal{G}}}^\dagger)$. Using the concentration of Gaussian vector, we have that with probability at least $1 - \beta$, $\mathsf{Tr}(M_1^\top \Sigma_{\widehat{\mathcal{G}}} M_1 \Sigma_{\widehat{\mathcal{G}}}^\dagger) = O(n)$. Similarly, for the second term, we know that $L_{\widehat{\mathcal{G}}} \succeq w^2 \mathbb{I}$, which in turn implies that $L_{\widehat{\mathcal{G}}}^\dagger \preceq w^{-2} \mathbb{I}$. Let $R \sim \mathcal{N}(0, 1)^{\binom{n}{2} \times n}$. We

$$
w^2 \mathsf{Tr}\left( M_2^\top M_2 L_{\widehat{\mathcal{G}}}^\dagger \right) = w^2 \mathsf{Tr}(\frac{1}{n} R^\top R L_{\widehat{\mathcal{G}}}^\dagger) \leq w^2 \frac{1}{n} \times \binom{n}{2} \times n \times \frac{1}{w^2} = \binom{n}{2}.
$$

Note that this defines the number of edges in $E_{\bar{\mathcal{G}}}^\top D E_{\bar{\mathcal{G}}}$; however, crucially, as we show later, we achieve multiplicative spectral approximation while preserving differential privacy. Both these points are crucial in achieving our final bounds as we use the fact that differential privacy is preserved under arbitrary post-processing by running the spectral sparsification algorithm on the private Laplacian $E_{\bar{\mathcal{G}}}^\top D E_{\bar{\mathcal{G}}}$ to reduce the number of edges to $O(n/\varepsilon^2)$ as claimed.

Let $e_i$ be the $i$-th row of $E_{\bar{\mathcal{G}}}$. Then

$$
\begin{aligned}
\widetilde{\tau}_i &= B_i L_{\widehat{\mathcal{G}}}^\dagger B_i^\top \\
&= e_i M_1 L_{\widehat{\mathcal{G}}}^\dagger M_1^\top e_i + w^2 M_2 L_{\widehat{\mathcal{G}}}^\dagger M_2^\top \\
&\geq \frac{1}{c} e_i L_{\bar{\mathcal{G}}}^\dagger e_i = \frac{1}{c} \tau_i,
\end{aligned}
$$

where the last inequality follows using the same argument as above. This completes the proof. $\square$

## B.4 Proof of Theorem 7

*Proof.* We first prove cut sparsification. Fix a set of nodes $S \subseteq [n]$. We have

$$
\mathsf{E}[\mathbf{1}_S^\top L_{\widehat{\mathcal{G}}} \mathbf{1}_S] = \alpha \mathbf{1}_S^\top L_{\mathcal{G}} \mathbf{1}, \qquad \mathsf{Var}(\mathbf{1}_S^\top L_{\widehat{\mathcal{G}}} \mathbf{1}_S) = \Theta(n(n - s)),
$$

where $s = |S|$. Using Chernoff bound,

$$
\mathsf{Pr}\left[ \left| \mathbf{1}_S^\top L_{\widehat{\mathcal{G}}} \mathbf{1}_S - \alpha \cdot \mathbf{1}_S^\top L_{\mathcal{G}} \mathbf{1}_S \right| \leq \sqrt{s(n - s) \log(2^n / \delta)} \right] \geq 1 - \delta/2^n.
$$

Using union bound over all $2^n$ possible choices of $S$, we have with probability $1 - \delta$,

$$\mathbf{1}_S^\top L_{\widehat{\mathcal{G}}} \mathbf{1}_S - \sqrt{n \mathbf{1}_S^\top L_n \mathbf{1}_S \log(1/\delta)} \leq \alpha \cdot \mathbf{1}_S^\top L_{\mathcal{G}} \mathbf{1}_S \leq \mathbf{1}_S^\top L_{\widehat{\mathcal{G}}} \mathbf{1}_S + \sqrt{n \mathbf{1}_S^\top L_n \mathbf{1}_S \log(1/\delta)}. \quad (3)$$

Using [35], we have $(1 - \varepsilon)L_{\widetilde{\mathcal{G}}} \preceq L_{\widehat{\mathcal{G}}} \preceq (1 + \varepsilon)L_{\widetilde{\mathcal{G}}}$ and $\widetilde{\mathcal{G}}$ has $O(n/\varepsilon^2)$ edges. Combining this with equation (3), cut sparsification follows. The privacy proof follows by simple case analysis. This completes the proof of Theorem 7. □

# C  Applications of Our Results

We presented differentially private algorithms for cut sparsifier and spectral sparsifier. In practice, cut sparsifier and spectral sparsifier are used as black-box to improve run-time efficiency by solving various graph related algorithms on sparse graphs instead of the original dense graph. For example, cut sparsifier can be used to compute approximations for maximum cut, sparsest cut, and maximum flow, etc. Since spectral sparsification also preserve spectral properties of the Laplacian of the graph, on top of cut related problems, they can be used to approximately solve linear systems over the Laplacian of G, and to approximate spectral clusterings, random walk properties, etc.

Our results can be applied to all these settings. This is because differential privacy is preserved under post-processing. Therefore, once we publish a differentially private representation of a graph, it can be used as black-box in any graph algorithm.

## C.1  Applications of Private Spectral Sparsification

Spectral sparsification is used as a first step for solving many graph related problems when the graph is dense and the complexity of the graph problems depends on the number of edges, such as cut queries. In many cases, it also allows one to approximately and efficiently infer many structural information about the graph that could be inferred using the spectral properties of the graph, for example, connectivity.

In what follows, we enumerate the implications of our results for answering all possible cut-queries, privately computing the set of vertices contributing in the approximate solution to MAX-CUT and SPARSE-CUT. All these results can also be achieved using cut-sparsification albeit with larger accuracy loss (see Appendix 5). We show the strength of private spectral sparsification over private cut sparsification by also privately computing Laplacian eigenmap, a popular approach used in manifold learning.

In addition to constructing private algorithms for many problems that have no prior such algorithms, our approach also leads to an efficient construction. This is because most of these applications runs in time dependent on the number of edges.

**Cut Queries.**  Spectral sparsifier also preserves all cuts in the graph. Consequently, the first application of spectral sparsifier is that we can compute all possible cut queries with error that scales linearly with $n$.

**Theorem 23.** *Given an $n$ vertices $m$ edges graph $\mathcal{G}$, there is an efficient $(\alpha, \beta)$-differentially private algorithm that preprocess the graph to output another graph $\widetilde{\mathcal{G}}$. Given a cut query in the form of set of vertices $S$, one can use $\widetilde{\mathcal{G}}$ to output the size of the cut with additive error $\tau \leq \widetilde{O}\left(\sqrt{\frac{s(n-s)}{\alpha^2}}\right)$ with probability at least $9/10$. Furthermore, answering a cut query, $S$, takes $O(|S|)$ amortized time.*

*Proof.*  We first present the algorithm and then prove that it achieves the bound claimed in the theorem. The algorithm is as follows:

**Algorithm.** Given a graph $\mathcal{G}$, compute a sparse graph $\widetilde{\mathcal{G}}$ using Theorem 39. Then given a vector $\mathbf{1}_S$, compute $\frac{1}{1 - \frac{w}{n}}\left(\mathbf{1}_S^\top L_{\widetilde{G}} \mathbf{1}_S - \frac{ws(n-s)}{n}\right)$.

We now argue the correctness of our algorithm. Using Theorem 39, we have

$$\mathbf{1}_S^\top L_{\widetilde{G}} \mathbf{1}_S \leq (1 + \varepsilon) \left( \left(1 - \frac{w}{n}\right) \Phi_S(\mathcal{G}) + \frac{w}{n} \mathbf{1}_S^\top L_n \mathbf{1}_S \right)$$

$$= (1 + \varepsilon) \left(1 - \frac{w}{n}\right) \Phi_S(\mathcal{G}) + (1 + \varepsilon) \frac{ws(n-s)}{n}$$

Therefore,

$$\frac{1}{1 - \frac{w}{n}} \left( \mathbf{1}_S^\top L_{\widetilde{G}} \mathbf{1}_S - \frac{ws(n-s)}{n} \right) \leq (1 + \varepsilon) \Phi_S(\mathcal{G}) + \frac{\varepsilon ws}{1 - \frac{w}{n}}$$

This completes the proof of Theorem 4. $\qquad\square$

Our algorithm answers a cut query, $S$, in $O(|S|)$ amortized time. This is a vast improvement over the previous best algorithm for answering cut queries [9, 54], which takes $O(n|S|)$ amortized cost. This improvement is especially significant when the graph is large, a more common phenomenon in practice. For example, social graphs like Facebook can have as many as $10^9$ vertices.

**MAX-CUT.** Another problem one can solve using our spectral sparsifier is that of MAX-CUT. Here, given a graph $\mathcal{G} = (V, E)$ on a vertex set $V$, the goal is to output a set of vertices $S \subseteq V$ that maximizes the value of $\Phi_S(\mathcal{G})$. It is well known that MAX-CUT is NP-hard. However, Goemans and Williamson gave an elegant polynomial time algorithm for computing $\zeta_{GW} - \eta$ approximation to MAX-CUT for an arbitrary constant $\eta > 0$, where

$$\zeta_{GW} := \min_{0 < \theta < \pi} \frac{\theta/\pi}{(1 - \cos\theta)/2} \approx 0.87856. \tag{4}$$

It is known that even approximation within a factor of $\zeta_{GW} + \rho$, for all $\rho > 0$ is NP-hard [31] under the unique games conjecture. [21] showed the following:

**Theorem 24** ( [21]). *Let $\eta > 0$ be an arbitrary small constant. For an $n$-vertex graph $\mathcal{G}$, there is a polynomial-time algorithm that produces a set of nodes $S$ satisfying $\Phi_S(\mathcal{G}) = (\zeta_{GW} - \eta)\mathsf{OPT}_{\max}$, where $\mathsf{OPT}_{\max}$ is the optimal max-cut.*

We show the following:

**Theorem 25.** *For an $n$-vertex graph $\mathcal{G} = (V, E)$, there is a polynomial-time algorithm that is $(\alpha, \beta)$-differentially private with respect to the edge level privacy and produces a set of nodes $S \subseteq V$ satisfying*

$$\Phi_S(\mathcal{G}) \geq (\zeta_{GW} - \eta) \left(\frac{1 - \varepsilon}{1 + \varepsilon}\right) \mathsf{OPT}_{\max} - O\left(\frac{|S|\sqrt{n}}{\alpha}\right)$$

*with probability at least $9/10$. Here $\zeta_{GW}$ is as defined in equation (4), $\mathsf{OPT}_{\max}$ is the optimal value of MAX-CUT and $\Phi_S(\mathcal{G})$ is the size of cut for vertex set in $S \subseteq V$.*

*Proof.* We first present the algorithm and then prove that it achieves the bound claimed in the theorem. The algorithm is as follows:

**Algorithm.** The algorithm first computes a private sparse graph using Algorithm 1, then runs the SDP based algorithm of [21] on the private sparse graph, and output the set of vertices $S$ outputted by that algorithm.

We next argue the correctness of our algorithm. Let $\widetilde{S}_{\mathsf{OPT}}$ be the solution of MAX-CUT on graph $\widetilde{\mathcal{G}}$ and $S_{\mathsf{OPT}}$ be the solution of MAX-CUT on the graph $\mathcal{G}$. In other words,

$$\widetilde{S}_{\mathsf{OPT}} = \operatorname*{argmax}_{S \subseteq V} \mathbf{1}_S^\top L_{\widetilde{\mathcal{G}}} \mathbf{1}_S \quad \text{and} \quad S_{\mathsf{OPT}} = \operatorname*{argmax}_{S \subseteq V} \mathbf{1}_S^\top L_{\mathcal{G}} \mathbf{1}_S.$$

From Theorem 24, we know that

$$\Phi_S(\widetilde{\mathcal{G}}) \geq (0.87856 - \eta) \max_{S \subseteq V} \mathbf{1}_S^\top L_{\widetilde{\mathcal{G}}} \mathbf{1}_S.$$

Further, since $\widetilde{\mathcal{G}}$ is a spectral sparsification, Theorem 3 implies that

$$(1-\varepsilon)\, x^\top \left( \frac{1}{\sqrt{n}} L_n + \left( 1 - \frac{1}{\sqrt{n}} \right) L_{\mathcal{G}} \right) x \le x^\top L_{\widetilde{\mathcal{G}}} x \le (1+\varepsilon)\, x^\top \left( \frac{1}{\sqrt{n}} L_n + \left( 1 - \frac{1}{\sqrt{n}} \right) L_{\mathcal{G}} \right) x.$$

holds simultaneously for $\forall x \in \mathbb{R}^n$ with probability $9/10$. Applying Theorem 3 twice, once with $\widetilde{S}_{\mathsf{OPT}}$ and $S_{\mathsf{OPT}}$, we have the desired claim. $\qquad\square$

**SPARSEST-CUT.** Another problem that is considered in graph theory is the problem of sparsest cut, SPARSE-CUT. Here, given a graph $\mathcal{G} = (V, E)$ on a vertex set $V$, the goal is to output a set of vertices $S$ that minimizes the value $\frac{\Phi_S(\mathcal{G})}{|S|(n-|S|)}$. For sparsest cut, [5] showed an $O(\sqrt{\log n})$ approximation.

**Theorem 26** ( [5]). *For an $n$-vertex graph $\mathcal{G}$, there is a polynomial-time algorithm that produces a set of nodes $S$ satisfying $\Phi_S(\mathcal{G}) = O(\sqrt{\log n})\mathsf{OPT}_{\mathsf{sparsest}}$, where $\mathsf{OPT}_{\mathsf{sparsest}}$ is the optimal sparsest cut.*

Combining Theorem 26 result with Theorem 3, we immediately get the following:

**Theorem 27.** *For an $n$-vertex graph $\mathcal{G} = (V, E)$, there is a polynomial-time algorithm that is $(\alpha, \beta)$-differentially private with respect to the edge level privacy and produces a set of nodes $S \subseteq V$ satisfying*

$$\Phi_S(\mathcal{G}) \le O(\sqrt{\log n}) \left( \frac{1+\varepsilon}{1-\varepsilon} \right) \mathsf{OPT}_{\mathsf{sparsest}} + O\left( \frac{\log^2 n}{\varepsilon\sqrt{n}} \right),$$

*with probability at least $9/10$. Here $\mathsf{OPT}_{\mathsf{sparsest}}$ is the optimal value of SPARSE-CUT and $\Phi_S(\mathcal{G})$ is the size of cut for vertex set in $S \subseteq V$.*

*Proof.* The algorithm is as follows:

**Algorithm.** The algorithm first computes a private sparse graph using Algorithm 1, then runs the SDP based algorithm of [5] on the private sparse graph, and output the set of vertices $S$ outputted by that algorithm.

We next argue the correctness of our algorithm. Let $\widetilde{S}_{\mathsf{OPT}}$ be the solution of SPARSEST-CUT on graph $\widetilde{\mathcal{G}}$ and $S_{\mathsf{OPT}}$ be the solution of SPARSEST-CUT on the graph $\mathcal{G}$. In other words,

$$\widetilde{S}_{\mathsf{OPT}} = \operatorname*{argmin}_{S \subseteq V} \left( \frac{\mathbf{1}_S^\top L_{\widetilde{\mathcal{G}}} \mathbf{1}_S}{|S|(n-|S|)} \right) \quad \text{and} \quad S_{\mathsf{OPT}} = \operatorname*{argmin}_{S \subseteq V} \left( \frac{\mathbf{1}_S^\top L_{\mathcal{G}} \mathbf{1}_S}{|S|(n-|S|)} \right).$$

From Theorem 26, we know that

$$\Phi_S(\widetilde{\mathcal{G}}) \le O(\sqrt{\log n}) \min_{S \subseteq V} \left( \frac{\mathbf{1}_S^\top L_{\widetilde{\mathcal{G}}} \mathbf{1}_S}{|S|(n-|S|)} \right).$$

Since $\widetilde{\mathcal{G}}$ is a spectral sparsification, Theorem 3 implies that

$$(1-\varepsilon)\, x^\top \left( \frac{1}{\sqrt{n}} L_n + \left( 1 - \frac{1}{\sqrt{n}} \right) L_{\mathcal{G}} \right) x \le x^\top L_{\widetilde{\mathcal{G}}} x \le (1+\varepsilon)\, x^\top \left( \frac{1}{\sqrt{n}} L_n + \left( 1 - \frac{1}{\sqrt{n}} \right) L_{\mathcal{G}} \right) x.$$

holds simultaneously for $\forall x \in \mathbb{R}^n$. Applying Theorem 3 twice, once with $\widetilde{S}_{\mathsf{OPT}}$ and $S_{\mathsf{OPT}}$, we have the desired claim. $\qquad\square$

**Edge Expansion.** The EDGE-EXPANSION ratio of a cut is the ratio of the weight of edges across the cut to the total weighted degree of edges incident to the side that is smaller with respect to total weighted degree. The edge expansion ratio of a graph is the minimum edge expansion ratio of any cut. [5] showed the following approximation result for edge expansion.

**Lemma 28** ( [5]). *For any constant $c > 0$ there is a polynomial-time algorithm that, given any regular weighted graph and a number $\gamma > 0$ behaves as follows. If the graphs has a $c$-balanced cut of edge expansion ratio less than $\gamma$ then the algorithm outputs a $c/2$-balanced cut of edge expansion ratio $2\sqrt{\gamma}$. If the graph does not have such a cut, the algorithm finds a set of at least $(1 - c/2)n$ nodes such that the induced subgraph (with deleted edges replaced by self-loops) on them has edge expansion ratio at least $2\gamma$.*

Using the fact that, up to a factor 2, computing the sparsest cut is the same as computing the *edge expansion* of the graph, we have the following corollary.

**Theorem 29.** *For an $n$-vertex graph $\mathcal{G} = (V, E)$, there is a polynomial-time algorithm that is $(\alpha, \beta)$-differentially private with respect to the edge level privacy and produces a set of nodes $S \subseteq V$ satisfying*

$$\mathsf{OUT} \leq O(\sqrt{\log n}) \left( \frac{1 + \varepsilon}{1 - \varepsilon} \right) \mathsf{OPT}_{\mathsf{edge}} + O \left( \frac{\log^2 n}{\varepsilon \sqrt{n}} \right),$$

*with probability at least $9/10$. Here $\mathsf{OPT}_{\mathsf{edge}}$ is the optimal value of* EDGE-EXPANSION.

## C.2 Laplacian Eigenmap

Recently, there has been renewed interest in the problem of developing low-dimensional representations when data arise from sampling a probability distribution on a manifold. This problem is referred to as manifold learning. There have been many approaches to manifold learning, such as *Isomap embedding*, *LLE embedding*, and *Laplacian eigenmap*. The locality-preserving character of the Laplacian eigenmap makes it relatively insensitive to outliers and noise, and hence the focus of this paper [5].

In the approach of Laplacian eigenmap, given $n$ data samples $\{x_1, \cdots, x_n\} \in \mathbb{R}^d$, we construct a weighted graph $\mathcal{G}$ with $n$ nodes and set of edges connecting neighboring points. The embedding map is now provided by computing the top $k$ eigenvectors of the graph Laplacian. There are multiple ways in which we assign an edge $e = (u, v)$ and edge weight between nodes $u$ and $v$. In this section, we consider an edge $e = (u, v)$ if $\|x_u - x_v\|_2 \leq \rho$ for some parameter $\rho$. If there is an edge, then that edge is given a weight, known as *heat kernel*, $e^{-\|x_u - x_v\|_2^2 / t}$ for some parameter $t \in \mathbb{R}$. The goal here is to find an embedding mapping, i.e., an orthonormal projection matrix, $UU^\top$, which is close to the optimal projection matrix $U_k U_k^\top$ (here columns of $U_k$ is formed by the top-$k$ eigenvectors of $L_{\mathcal{G}}$).

**Theorem 30.** *There is an efficient learning algorithm that $(\alpha, \beta)$-differentially privately output a rank-$k$ subspace $\widehat{P}$ such that*

$$\left\| (\mathbb{I} - \widehat{P}) L_{\mathcal{G}} \right\|_2 \leq \|L_{\mathcal{G}} - L_{\mathcal{G}, k}\|_2 + O \left( \sqrt{\frac{n \log(1/\beta)}{\alpha^2 \varepsilon^2}} \log \left( \frac{1}{\beta} \right) \right).$$

*Here $L_{\mathcal{G}, k} = \arg\min_X \|L_{\mathcal{G}} - X\|_2$ is the optimal subspace of $L_{\mathcal{G}}$.*

*Proof.* The algorithm is as follows:

**Algorithm.** Run the algorithm PRIVATE-SPARSIFY to output a Laplacian $L_{\mathsf{priv}}$. Compute the top-$k$ singular vectors of $L_{\mathsf{priv}}$ and output it.

We next argue the correctness of our algorithm to compute Laplacian eigenmap. We first fix some notations. Let $\widehat{P}$ be the top-$k$ singular vectors of $L_{\mathsf{priv}}$ and $\widetilde{P}$ be the top-$k$ singular vectors of $L_{\widehat{\mathcal{G}}}$. We use the notation $\sigma(A)$ to denote the spectral norm of the matrix $A$. Denote by $\widehat{Q} = \mathbb{I} - \widehat{P}, \widetilde{Q} = \mathbb{I} - \widetilde{P}$. Let $\widetilde{x}$ be the witness vector for $\widetilde{P}$ and $\widehat{x}$ be a witness vector for $\widehat{P}$. Further, let

$$\widetilde{x}_{L_{\mathsf{priv}}} = \underset{x, \|x\|_2 = 1}{\arg\max} \; x^\top \widetilde{Q}^\top L_{\mathsf{priv}} \widetilde{Q} x, \quad \widehat{x}_{L_{\widehat{\mathcal{G}}}} = \underset{x, \|x\|_2 = 1}{\arg\max} \; x^\top \widehat{Q}^\top L_{\widehat{\mathcal{G}}} \widehat{Q} x_2.$$

Using the definition of $\widehat{x}, \widetilde{x}_{L_{\mathsf{priv}}}$, and $\widetilde{Q}$, we have

$$\widehat{x}^\top \widehat{Q}^\top L_{\mathsf{priv}} \widehat{Q} \widehat{x} = \sigma \left( \widehat{Q} L_{\mathsf{priv}} \widehat{Q} \right) \leq \sigma \left( \widetilde{Q} L_{\mathsf{priv}} \widetilde{Q} \right) = \widetilde{x}_{L_{\mathsf{priv}}}^\top \widetilde{Q}^\top L_{\mathsf{priv}} \widetilde{Q} \widetilde{x}_{L_{\mathsf{priv}}}, \tag{5}$$

where the inequality is due to the fact that $\widehat{Q}$ mininimizes the quantity on the left.

Then we have the following:

$$
\begin{aligned}
\sigma\left(\widetilde{Q}L_{\widehat{\mathcal{G}}}\widetilde{Q}\right) &= \widetilde{x}^{\top}\widetilde{Q}^{\top}L_{\widehat{\mathcal{G}}}\widetilde{Q}\widetilde{x} \\
&\geq \widetilde{x}_{L_{\mathsf{priv}}}^{\top}\widetilde{Q}^{\top}L_{\widehat{\mathcal{G}}}\widetilde{Q}\widetilde{x}_{L_{\mathsf{priv}}} && \text{(Maximality)} \\
&\geq (1-\varepsilon)\widetilde{x}_{L_{\mathsf{priv}}}^{\top}\widetilde{Q}^{\top}L_{\mathsf{priv}}\widetilde{Q}\widetilde{x}_{L_{\mathsf{priv}}} && \text{(Spectral guarantee)} \\
&\geq (1-\varepsilon)\widehat{x}^{\top}\widehat{Q}^{\top}L_{\mathsf{priv}}\widehat{Q}\widehat{x} && \text{(equation (5))} \\
&= (1-\varepsilon)\sigma\left(\widehat{Q}L_{\mathsf{priv}}\widehat{Q}\right).
\end{aligned}
$$

Similarly, we have the following:

$$
\begin{aligned}
\sigma\left(\widehat{Q}L_{\mathsf{priv}}\widehat{Q}\right) &= \widehat{x}^{\top}\widehat{Q}^{\top}L_{\mathsf{priv}}\widehat{Q}\widehat{x} \\
&\geq \widehat{x}_{L_{\widehat{\mathcal{G}}}}^{\top}\widehat{Q}^{\top}L_{\mathsf{priv}}\widehat{Q}\widehat{x}_{L_{\widehat{\mathcal{G}}}} && \text{(Maximality)} \\
&\geq \frac{1}{(1+\varepsilon)}\widehat{x}_{L_{\widehat{\mathcal{G}}}}^{\top}\widehat{Q}^{\top}L_{\widehat{\mathcal{G}}}\widehat{Q}\widehat{x}_{L_{\widehat{\mathcal{G}}}} && \text{(Spectral guarantee)} \\
&= \frac{1}{(1+\varepsilon)}\sigma\left(\widehat{Q}L_{\widehat{\mathcal{G}}}\widehat{Q}\right). && \text{(By definition of } \widehat{x}_{L_{\widehat{\mathcal{G}}}}\text{)}
\end{aligned}
$$

In other words,

$$
\sigma\left(\widehat{Q}L_{\widehat{\mathcal{G}}}\widehat{Q}\right) \leq \frac{1+\varepsilon}{1-\varepsilon}\sigma\left(\widetilde{Q}L_{\widehat{\mathcal{G}}}\widetilde{Q}\right)
$$

Since $L_{\widehat{\mathcal{G}}} = \left(1-\frac{w}{n}\right)L_{\mathcal{G}} + \frac{w}{n}L_n$, subadditivity of spectral norms implies

$$
\sigma\left(\widehat{Q}L_{\widehat{\mathcal{G}}}\widehat{Q}\right) \geq \sigma(\widehat{Q}L_{\mathcal{G}}\widehat{Q}) - \sigma\left(\frac{w}{n}L_n\right) = \sigma(\widehat{Q}L_{\mathcal{G}}\widehat{Q}) - w \quad \text{and}
$$

$$
\sigma\left(\widetilde{Q}L_{\widehat{\mathcal{G}}}\widetilde{Q}\right) = \sigma\left(L_{\widehat{\mathcal{G}}} - L_{\widehat{\mathcal{G}},k}\right) \leq \sigma\left(L_{\mathcal{G}} - L_{\widehat{\mathcal{G}},k}\right) + \sigma\left(\frac{w}{n}L_n\right) = \sigma\left(L_{\mathcal{G}} - L_{\widehat{\mathcal{G}},k}\right) + w.
$$

Using Aclioptas and McSherry [1], we know that

$$
\left\|L_{\mathcal{G}} - L_{\widehat{\mathcal{G}},k}\right\|_2 \leq \|L_{\mathcal{G}} - L_{\mathcal{G},k}\|_2 + O(\sqrt{n}).
$$

This implies that

$$
\left\|(\mathbb{I} - \widehat{P})L_{\mathcal{G}}\right\|_2 \leq \frac{1+\varepsilon}{1-\varepsilon}\|L_{\mathcal{G}} - L_{\mathcal{G},k}\|_2 + O(w)
$$

The theorem follows after rescaling the value of $\varepsilon$. $\qquad\square$

## C.3 Applications of Cut Sparsification: Achieving Differential Privacy

Cut sparsifiers were introduced by [8] to compute approximations for maximum cut, sparsest cut, and maximum flow, etc. Our results can be applied to all these settings. The main benefit of using cut-sparsifier is that we get $(\alpha, 0)$-differential privacy with a comparatively more efficient algorithm.

**Cut Queries.** The first application of cut sparsifier is that we can efficiently compute all possible cut queries with error that scales linearly with $n$.

**Theorem 31.** *Given an $n$ vertices $m$ edges graph $\mathcal{G}$, there is an efficient $\alpha$-differentially private algorithm that preprocess the graph in polynomial time to output another graph $\widetilde{\mathcal{G}}$. Further, given a cut query in the form of set of vertices $S$, one can output the cut with additive error $\tau \leq \widetilde{O}\left(\sqrt{\frac{sn(n-s)}{\alpha^2}}\right)$ with probability at least $99/100$. Further, computing a cut query takes $O(s)$ time.*

*Proof.* The algorithm is as follows:

**Algorithm.** Given a graph $\mathcal{G}$, compute a sparse graph $\widetilde{\mathcal{G}}$ using Theorem 7. Then given a vector $\mathbf{1}_S$, compute $\alpha^{-1}\mathbf{1}_S^\top L_{\widetilde{G}}\mathbf{1}_S$.

We next argue the correctness of the algorithm. Using Theorem 7, we have

$$\frac{1}{\alpha}\mathbf{1}_S^\top L_{\widetilde{G}}\mathbf{1}_S \le (1+\varepsilon)\Phi_S(\mathcal{G}) + \frac{\sqrt{n\mathbf{1}_S^\top L_n\mathbf{1}_S}}{\alpha}$$
$$= (1+\varepsilon)\Phi_S(\mathcal{G}) + O\left(\sqrt{\frac{sn(n-s)}{\alpha^2}}\right)$$

with probability at least $99/100$. Similarly, for the lower bound, we have

$$\frac{1}{\alpha}\mathbf{1}_S^\top L_{\widetilde{G}}\mathbf{1}_S \ge (1-\varepsilon)\Phi_S(\mathcal{G}) - \frac{\sqrt{n\mathbf{1}_S^\top L_n\mathbf{1}_S}}{\alpha}$$
$$= (1-\varepsilon)\Phi_S(\mathcal{G}) - O\left(\sqrt{\frac{sn(n-s)}{\alpha^2}}\right)$$

This completes the proof of Theorem 31. $\qquad\qquad\qquad\qquad\qquad\qquad\qquad\square$

**MAX-CUT.** Another problem one can solve using our cut sparsifier is that of MAX-CUT. Here, given a graph $\mathcal{G} = (V, E)$ on vertex set $V$, the goal is to output a set of vertices $S \subseteq V$ that maximizes the value of $\Phi_S(\mathcal{G})$. It is well known that MAX-CUT is NP-hard. However, Goemans and Williamson gave an elegant polynomial time algorithm for computing $\zeta_{GW} - \eta$ approximation to MAX-CUT—it is known that even approximation within a factor of $\zeta_{GW} + \rho$, for all $\rho > 0$ is NP-hard [31] under the unique games conjecture [30], where $\zeta_{GW}$ is as defined in equation (4).

**Theorem 32.** *For an $n$-vertex graph $\mathcal{G} := (V, E)$, there is a polynomial-time algorithm that is $(\alpha, 0)$-differentially private with respect to the edge level privacy and produces a set of nodes $S \subseteq V$ satisfying*

$$\Phi_S(\mathcal{G}) \ge (\zeta_{GW} - \eta)\left(\frac{1-\varepsilon}{1+\varepsilon}\right)\mathsf{OPT}_{\mathsf{max}} - O\left(\frac{sn}{\alpha}\right),$$

*with probability at least $9/10$. Here $\zeta_{GW}$ is as defined in equation (4), $\mathsf{OPT}_{\mathsf{max}}$ is the optimal value of MAX-CUT and $\Phi_S(\mathcal{G})$ is the size of cut for vertex set in $S \subseteq V$.*

*Proof.* The idea is to first compute a private sparse graph using Algorithm 2, then run the SDP based algorithm of [21] on the private sparse graph, and output the set of vertices $S$ outputted by that algorithm. Let $\widetilde{S}_{\mathsf{OPT}}$ be the solution of MAX-CUT on graph $\widetilde{\mathcal{G}}$ and $S_{\mathsf{OPT}}$ be the solution of MAX-CUT on the graph $\mathcal{G}$. In other words,

$$\widetilde{S}_{\mathsf{OPT}} = \operatorname*{argmax}_{S \subseteq V} \mathbf{1}_S^\top L_{\widetilde{\mathcal{G}}}\mathbf{1}_S \quad \text{and} \quad S_{\mathsf{OPT}} = \operatorname*{argmax}_{S \subseteq V} \mathbf{1}_S^\top L_{\mathcal{G}}\mathbf{1}_S.$$

From Theorem 24, we know that

$$\Phi_S(\widetilde{\mathcal{G}}) \ge (\zeta_{GW} - \eta)\max_{S \subseteq V} \mathbf{1}_S^\top L_{\widetilde{\mathcal{G}}}\mathbf{1}_S$$

with probability at least $99/100$. Further, since $\widetilde{\mathcal{G}}$ is a cut-sparsification, Theorem 7 implies that

$$(1-\varepsilon)\mathbf{1}_S^\top L_{\widetilde{\mathcal{G}}}\mathbf{1}_S - O\left(\sqrt{n\mathbf{1}_S^\top L_n\mathbf{1}_S}\right) \le \alpha\mathbf{1}_S^\top L_{\mathcal{G}}\mathbf{1}_S \le (1+\varepsilon)\mathbf{1}_S^\top L_{\widetilde{\mathcal{G}}}\mathbf{1}_S + O\left(\sqrt{n\mathbf{1}_S^\top L_n\mathbf{1}_S}\right)$$

holds simultaneously for $\forall S \subseteq [n]$ with probability $99/100$. Applying Theorem 7 twice, once with $\widetilde{S}_{\mathsf{OPT}}$ and $S_{\mathsf{OPT}}$, we have the desired claim. $\qquad\qquad\qquad\qquad\qquad\square$

**SPARSEST-CUT.** Another problem that is considered in graph theory is the problem of sparsest cut, SPARSE-CUT. Here, given a graph $\mathcal{G} = (V, E)$ on vertex set $V$, the goal is to output a set of vertices $S$ that minimizes the value $\frac{\Phi_S(\mathcal{G})}{|S|(n-|S|)}$.

Combining Theorem 26 result with Theorem 7, we immediately get the following:

**Theorem 33.** *For an n-vertex graph $\mathcal{G} := (V, E)$, there is a polynomial-time algorithm that is $(\alpha, 0)$-differentially private with respect to the edge level privacy and produces a set of nodes $S$ satisfying*

$$\Phi_S(\mathcal{G}) \leq O(\sqrt{\log n}) \left( \frac{1+\varepsilon}{1-\varepsilon} \right) \mathsf{OPT}_{\mathsf{sparsest}} + O\left( \frac{\log^2 n}{\varepsilon\sqrt{|S|}} \right),$$

*with probability at least $9/10$. Here $\mathsf{OPT}_{\mathsf{sparsest}}$ is the optimal value of SPARSE-CUT and $\Phi_S(\mathcal{G})$ is the size of cut for vertex set in $S \subseteq V$.*

*Proof.* The idea is to first compute a private sparse graph using Algorithm 2, then run the SDP based algorithm of [5] on the private sparse graph, and output the set of vertices $S$ outputted by that algorithm. Let $\widetilde{S}_{\mathsf{OPT}}$ be the solution of SPARSEST-CUT on graph $\widetilde{\mathcal{G}}$ and $S_{\mathsf{OPT}}$ be the solution of SPARSEST-CUT on the graph $\mathcal{G}$. In other words,

$$\widetilde{S}_{\mathsf{OPT}} = \underset{S \subseteq V}{\mathrm{argmin}} \left( \frac{\mathbf{1}_S^\top L_{\widetilde{\mathcal{G}}} \mathbf{1}_S}{|S|(n-|S|)} \right) \quad \text{and} \quad S_{\mathsf{OPT}} = \underset{S \subseteq V}{\mathrm{argmin}} \left( \frac{\mathbf{1}_S^\top L_{\mathcal{G}} \mathbf{1}_S}{|S|(n-|S|)} \right).$$

From Theorem 26, we know that

$$\Phi_S(\widetilde{\mathcal{G}}) \leq O(\sqrt{\log n}) \min_{S \subseteq V} \left( \frac{\mathbf{1}_S^\top L_{\widetilde{\mathcal{G}}} \mathbf{1}_S}{|S|(n-|S|)} \right)$$

with probability at least $99/100$. Further, since $\widetilde{\mathcal{G}}$ is a cut-sparsification, Theorem 7 implies that

$$(1-\varepsilon)\mathbf{1}_S^\top L_{\widetilde{\mathcal{G}}} \mathbf{1}_S - O\left( \sqrt{n\mathbf{1}_S^\top L_n \mathbf{1}_S} \right) \leq \alpha \mathbf{1}_S^\top L_{\mathcal{G}} \mathbf{1}_S \leq (1+\varepsilon)\mathbf{1}_S^\top L_{\widetilde{\mathcal{G}}} \mathbf{1}_S + O\left( \sqrt{n\mathbf{1}_S^\top L_n \mathbf{1}_S} \right)$$

holds simultaneously for $\forall S \subseteq V$ with probability at least $99/100$. Applying Theorem 7 twice, once with $\widetilde{S}_{\mathsf{OPT}}$ and $S_{\mathsf{OPT}}$, we have the desired claim. $\square$

**EDGE-EXPANSION.** Using the fact that, up to a factor 2, computing the sparsest cut is the same as computing the *edge expansion* of the graph, we have the following corollary.

**Theorem 34.** *For an n-vertex graph $\mathcal{G} = (V, E)$, there is a polynomial-time algorithm that is $(\alpha, \beta)$-differentially private with respect to the edge level privacy and produces a set of nodes $S \subseteq V$ satisfying*

$$\mathsf{OUT} \leq O(\sqrt{\log n}) \left( \frac{1+\varepsilon}{1-\varepsilon} \right) \mathsf{OPT}_{\mathsf{edge}} + O\left( \frac{\log^2 n}{\varepsilon\sqrt{|S|}} \right),$$

*with probability at least $9/10$. Here $\mathsf{OPT}_{\mathsf{edge}}$ is the optimal value of EDGE-EXPANSION.*

# D  Flexibility of Our Approach

In this section, we illustrate the flexibility of our approach. For the ease of presentation, we first assume that the graph is unweighted. Later, in Section D.1, we show how to remove this assumption. We prove Theorem 35 through a series of lemmata. We first show in Lemma 36 that we can privately compute an overestimate of the leverage scores and still the vector of leverage scores has a small $\ell_0$ norm. This implies that the number of edges in our sparsified graph is of order $\widetilde{O}(n/\varepsilon^2)$.

**Theorem 35.** *Let $\mathcal{G}$ be a graph on $n$ vertices. Given an approximation parameter $0 < \varepsilon < 1$, confidence parameter $\delta$, and privacy parameters $\alpha, \beta$, let $w = O\left( \sqrt{\frac{n\log(1/\beta)}{\alpha^2\varepsilon^2}} \log(1/\beta) \right)$. Then PRIVATE-SPARSIFY-ADD-MULT, described in Algorithm 3, is $(\alpha, \beta)$-differentially private algorithm,*

---

**Algorithm 3** PRIVATE-SPARSIFY-ADD-MULT $(\mathcal{G}, \varepsilon, (\alpha, \beta))$

---

**Input:** An $n$ vertex graph $\mathcal{G} = (V, E)$, privacy parameters $(\alpha, \beta)$, approximation parameter $\varepsilon$.

**Output:** A Laplacian $L_{\widetilde{\mathcal{G}}}$.

1: **Initialization.** Sample a random Gaussian matrix $M \in \mathbb{R}^{n/\varepsilon^2 \times m}$ such that $M_{ij} \sim \mathcal{N}(0, \varepsilon^2/n)$.

2: **Privatize.** Compute a graph $\widehat{\mathcal{G}}$ with Laplacian $L_{\widehat{\mathcal{G}}} = \left(1 - \frac{w}{n}\right) L_{\mathcal{G}} + \frac{w}{n} L_n$, where $w = O\left(\sqrt{\frac{n \log(1/\beta)}{\alpha^2 \varepsilon^2}} \log\left(\frac{1}{\beta}\right)\right)$. Let $E_{\widehat{\mathcal{G}}}$ be the corresponding edge-adjacency matrix and $E_{K_n}$ be the edge-adjacency matrix of weighted complete graph, $\sqrt{w/n} K_n$. Compute $H = M E_{\widehat{\mathcal{G}}}$.

3: **Compute effective resistance.** $\widetilde{\tau}_i = e_i (H^\top H)^\dagger e_i^\top$, where $e_i$ is $i$-th the row of $E_{K_n}$.

4: **Construct** diagonal matrix $D \in \mathbb{R}^{m \times m}$ whose diagonal entries are $D_{ii} := p_i^{-1}$ with probability $p_i$, where $p_i = \min\left\{c\widetilde{\tau}_i \varepsilon^{-2} \log(n/\delta), 1\right\}$, and 0 otherwise.

5: **Output.** $L_{\widetilde{\mathcal{G}}} := E_{K_n}^\top D E_{K_n}$.

---

*and outputs a Laplacian $L_{\widetilde{\mathcal{G}}}$ of $\widetilde{O}(n/\varepsilon^2)$ edges graph, such that, with probability at least $99/100$, we have*

$$\widetilde{O}\left(\frac{1-\varepsilon}{\sqrt{n}}\right) L_n - \varepsilon\left(1 - \frac{w}{n}\right) L_{\mathcal{G}} \preceq L_{\widetilde{\mathcal{G}}} \preceq \varepsilon\left(1 - \frac{w}{n}\right) L_{\mathcal{G}} + \widetilde{O}\left(\frac{1+\varepsilon}{\sqrt{n}}\right) L_n,$$

*where $L_{\mathcal{G}}$ is the Laplacian of the input graph $\mathcal{G}$ and $L_n$ is the Laplacian of complete graph $K_n$.*

We first give a proof sketch. The privacy proof follows from [9, 54]. To complete the proof, we prove the following: $\tau_i \leq \widetilde{\tau}_i$, where $\tau_i$ are the true effective resistance, $\sum_i \widetilde{\tau}_i = O(n)$, and the spectral guarantee. Since $M$ is a random Gaussian matrix, we can show that $(1 - \varepsilon) L_{\widehat{\mathcal{G}}} \preceq E_{\widehat{\mathcal{G}}}^\top M^\top M E_{\widehat{\mathcal{G}}} \preceq (1 + \varepsilon) L_{\widehat{\mathcal{G}}}$. Intuitively, our sampling is equivalent to sampling by the standard effective resistance of $L_{\widehat{\mathcal{G}}}$. We use the matrix Bernstein to show that sampling by these effective resistance will yield $L_{\widetilde{\mathcal{G}}}$ satisfying $(1 - \varepsilon) L_{\widehat{\mathcal{G}}} \preceq L_{\widetilde{\mathcal{G}}} \preceq (1 + \varepsilon) L_{\widehat{\mathcal{G}}}$. However, we actually sample the edges of complete graph $K_n$. Subtracting off the effect of $\left(1 - \frac{w}{n}\right) L_{\mathcal{G}}$ yields the mixed additive-multiplicative bound. We now give a detailed proof.

**Lemma 36.** *Let $M$ be an $O\left(\frac{n \log(1/\delta)}{\varepsilon^2}\right) \times \binom{n}{2}$ random Gaussian matrix with entries sampled iid from $\mathcal{N}(0, 2\varepsilon/n \log(1/\delta))$. Let $E_{\widehat{\mathcal{G}}}$ be the edge-adjacency matrix of the graph $\widehat{\mathcal{G}}$ formed by overlaying a weighted complete graph $\frac{w}{n} L_n$ on top of the input graph, i.e., $L_{\widehat{\mathcal{G}}} = \left(1 - \frac{w}{n}\right) L_{\mathcal{G}} + \frac{w}{n} K_n$. Let $e_i$ be the $i$-th row for edge-adjacency matrix of complete graph $K_n$. Define a diagonal matrix $\widetilde{\tau} \in \mathbb{R}^{\binom{n}{2} \times \binom{n}{2}}$*

$$\widetilde{\tau}_i = e_i \left(E_{\widehat{\mathcal{G}}}^\top M^\top M E_{\widehat{\mathcal{G}}}\right)^\dagger e_i^\top.$$

*Then $\|\widetilde{\tau}\|_0 \leq \widetilde{O}(n(1+\varepsilon)/\varepsilon^2)$.*

*Proof.* Let $E_{\widehat{\mathcal{G}}}^\top E_{\widehat{\mathcal{G}}} = U \Lambda^2 U^\top$. Since $L_{\widehat{\mathcal{G}}}$ has rank $n - 1$ and $M$ is a random Gaussian matrix of dimensions $O\left(\frac{n \log(1/\delta)}{\varepsilon^2}\right) \times \binom{n}{2}$, [48] gives us that with probability $1 - \delta$,

$$(1 - \varepsilon) L_{\widehat{\mathcal{G}}} \preceq E_{\widehat{\mathcal{G}}}^\top M^\top M E_{\widehat{\mathcal{G}}} \preceq (1 + \varepsilon) L_{\widehat{\mathcal{G}}}$$

In other words, $(1 - \varepsilon) L_{\widehat{\mathcal{G}}}^\dagger \preceq (E_{\widehat{\mathcal{G}}}^\top M^\top M E_{\widehat{\mathcal{G}}})^\dagger \preceq (1 + \varepsilon) L_{\widehat{\mathcal{G}}}^\dagger$. This in particular implies that $\widetilde{\tau}_i \leq (1 + \varepsilon) \tau_i$, where $\tau_i$ is the true effective resistance for $i$-th edge.

Set $p_i = \min\left\{\widetilde{\tau}_{i,i} \frac{c \log(n/\delta)}{\varepsilon^2}, 1\right\}$ for $1 \leq i \leq n$. Let $D$ is a diagonal matrix whose diagonal entries are $p_i^{-1}$ with probability $p_i$. The expected number of edges in the resulting graph is $\sum_i \widetilde{\tau}_i$. So Chernoff

bound will give that with probability $1 - \delta$, the number of edges in the graph $\widetilde{\mathcal{G}}$ is

$$
\begin{aligned}
\sum_i p_i &\leq \frac{c\log(n/\delta)}{\varepsilon^2} \sum_i \widetilde{\tau}_i \\
&\leq \frac{c(1+\varepsilon)\log(n/\delta)}{\varepsilon^2} \operatorname{Tr}\left(E_{K_n} L_{\widehat{\mathcal{G}}}^{\dagger} E_n^{\top}\right) \\
&= \frac{c(1+\varepsilon)\log(n/\delta)}{\varepsilon^2} \operatorname{Tr}\left(E_n^{\top} E_{K_n} L_{\widehat{\mathcal{G}}}^{\dagger}\right) \\
&\leq \frac{c(1+\varepsilon)\log(n/\delta)}{\varepsilon^2} \left(\operatorname{Tr}\left(E_n^{\top} E_{K_n} L_{\widehat{\mathcal{G}}}^{\dagger}\right) + \operatorname{Tr}\left(E_{\mathcal{G}}^{\top} E_{\mathcal{G}} L_G^{\dagger}\right)\right) \\
&= \frac{c(1+\varepsilon)\log(n/\delta)}{\varepsilon^2} \operatorname{Tr}\left(E_{\widehat{\mathcal{G}}} L_{\widehat{\mathcal{G}}}^{\dagger} E_{\widehat{\mathcal{G}}}^{\top}\right) \\
&\leq \frac{cn(1+\varepsilon)\log(n/\delta)}{\varepsilon^2}.
\end{aligned}
$$

This completes the proof of Lemma 36. $\qquad\square$

Equipped with Lemma 36, we can now prove the spectral sparsification guarantee of our output.

**Lemma 37.** *With probability at least $1 - \delta$, we have*

$$
\widetilde{O}\left(\frac{1-\varepsilon}{\sqrt{n}}\right) L_n - \varepsilon\left(1 - \frac{w}{n}\right) L_{\mathcal{G}} \preceq \varepsilon\left(1 - \frac{w}{n}\right) L_{\mathcal{G}} + L_{\widetilde{\mathcal{G}}} \preceq \widetilde{O}\left(\frac{1+\varepsilon}{\sqrt{n}}\right) L_n
$$

*Proof.* We consider the graph $\mathcal{G}$ overlaid with a weighted complete graph with weights $w/n$, i.e.,

$$
L_{\widehat{\mathcal{G}}} = \frac{w}{n} L_n + \left(1 - \frac{w}{n}\right) L_{\mathcal{G}}.
$$

In other words, we have $E_{K_n}^{\top} D E_{K_n}$. More precisely, $\tau_i = \frac{w}{n} e_i L_{\widehat{\mathcal{G}}}^{\dagger} e_i^{\top} = \frac{w}{n} e_i U_{\widehat{\mathcal{G}}} \Sigma_{\widehat{\mathcal{G}}}^{-2} U_{\widehat{\mathcal{G}}}^{\top} e_i^{\top}$. We borrow the idea from [14]. Let define $Q_i = \sqrt{\frac{w}{n}} e_i U_{\widehat{\mathcal{G}}} \Sigma_{\widehat{\mathcal{G}}}^{-1}$. Next define the following matrix valued random variable:

$$
X_i := \begin{cases} \left(\frac{1}{p_i} - 1\right) Q_i^{\top} Q_i & \text{with probability } p_i \\ -Q_i^{\top} Q_i & \text{with probability } 1 - p_i \end{cases}.
$$

Now $Y = \sum_i X_i$. Then we have

$$
\mathsf{E}[Y] = \sum_{i=1}^{n} \left(p_i \left(\frac{1}{p_i} - 1\right) Q_i^{\top} Q_i - (1 - p_i) Q_i^{\top} Q_i\right) = 0.
$$

By the definition of $p_i$, $\|X_i\|_2 = 1$. We have to bound $\sigma^2 = \left\|\mathsf{E}\left[Y^2\right]\right\|_2$. For this note that

$$
\begin{aligned}
\mathsf{E}[Y^2] &= \sum_i \left[p_i \left(\frac{1}{p_i} - 1\right)^2 + (1 - p_i)\right] \frac{w^2}{n^2} (e_i U_{\widehat{\mathcal{G}}} \Sigma_{\widehat{\mathcal{G}}}^{-1})^{\top} e_i U_{\widehat{\mathcal{G}}} \Sigma_{\widehat{\mathcal{G}}}^{-2} U_{\widehat{\mathcal{G}}}^{\top} e_i^{\top} e_i U_{\widehat{\mathcal{G}}} \Sigma_{\widehat{\mathcal{G}}}^{-1} \\
&\preceq \sum_i O\left(\frac{\tau_i}{p_i}\right) \frac{w}{n} \Sigma_{\widehat{\mathcal{G}}}^{-1} U_{\widehat{\mathcal{G}}}^{\top} e_i^{\top} e_i U_{\widehat{\mathcal{G}}} \Sigma_{\widehat{\mathcal{G}}}^{-1} \\
&\preceq O\left(\frac{\varepsilon^2}{\log(n/\delta)}\right) \Sigma_{\widehat{\mathcal{G}}}^{-1} U_{\widehat{\mathcal{G}}}^{\top} \left(\frac{w}{n} L_n\right) U_{\widehat{\mathcal{G}}} \Sigma_{\widehat{\mathcal{G}}}^{-1} \\
&\preceq O\left(\frac{\varepsilon^2}{\log(n/\delta)}\right) \mathbb{I}.
\end{aligned}
$$

This is because $\frac{w}{n} L_n \preceq U_{\widehat{\mathcal{G}}} \Sigma_{\widehat{\mathcal{G}}}^2 U_{\widehat{\mathcal{G}}}^{\top}$. In other words, $\sigma^2 \leq O\left(\frac{\varepsilon^2}{\log(n/\delta)}\right)$. Further, we can compute the intrinsic dimension as

$$
\frac{\operatorname{Tr}(\mathbb{I})}{\|\mathbb{I}\|_2} = n - 1.
$$

Using Matrix Bernstein inequality for intrinsic dimension [53] now gives us the following:

$$\Pr\left[\|Y\|_2 \geq \varepsilon\right] \leq 4n e^{-c\ln(n/\delta)/2} \leq \delta/2$$

for large enough $c$. Noting that $L_{\widetilde{\mathcal{G}}} = U_{\widehat{G}}\Sigma_{\widehat{G}}Y\Sigma_{\widehat{G}}U_{\widehat{G}}^\top + \frac{w}{n}L_n$ gives us that

$$\widetilde{O}\left(\frac{1-\varepsilon}{\sqrt{n}}\right)L_n - \varepsilon\left(1 - \frac{w}{n}\right)L_{\mathcal{G}} \preceq L_{\widetilde{\mathcal{G}}} \preceq \varepsilon\left(1 - \frac{w}{n}\right)L_{\mathcal{G}} + \widetilde{O}\left(\frac{1+\varepsilon}{\sqrt{n}}\right)L_n$$

This completes the proof of Lemma 37. $\qquad\square$

**Lemma 38.** $L_{\widetilde{\mathcal{G}}}$ *outputted by* PRIVATE-SPARSIFY-ADD-MULT *is* $(\alpha, \beta)$-*differentially private with respect to the edge-level privacy.*

*Proof.* We first prove the privacy guarantee. The only time the graph is used in the algorithm PRIVATE-SPARSIFY-ADD-MULT is when we compute $H := ME_{\widehat{G}}$. This is differentially private due to [9, 54] by our choice of $w$. Using the post-processing property (Lemma 15) of differential privacy, the result follows. $\qquad\square$

### D.1 Extension to Weighted Graph

We can use a standard technique to extend our result for Theorem 35 to weighted graphs in which an edge's weight is specified. We assume that the weights on the graph are integers in the range $[1, \text{poly } n]$. We consider different levels $(1 + \varepsilon)^i$ for $i \in [c \log n]$ for some constant $c$. Then we consider input graphs being partitioned in form

$$L_{\mathcal{G}} = \sum_{i=1}^{c\log n} L_{\mathcal{G},i},$$

where $L_{\mathcal{G},i}$ has edges with weights $\{0, (1+\varepsilon)^i\}$. In other words, we use the $(1+\varepsilon)$-ary representation of weights on the edges and partition the graph accordingly. Again since there are at most poly $\log n$ levels, the number of edges in $\widetilde{\mathcal{G}}$ is $\widetilde{O}(n/\varepsilon^2)$. Since $\widetilde{\mathcal{G}}$ is $(\alpha, \beta)$-differentially private, we can run another instance of [35] to get $O(n/\varepsilon^2)$ edge graph $\widehat{\mathcal{G}}$. Using Lemma 22, we therefore have

**Theorem 39.** *Given the privacy parameter* $(\alpha, \beta)$, *the accuracy parameter* $\varepsilon$ *and confidence parameter* $\delta$, *let* $w = \widetilde{O}\left(\sqrt{\frac{n\log(1/\beta)}{\alpha^2\varepsilon^2}}\log\left(\frac{1}{\beta}\right)\right)$ . *Given a weighted graph* $\mathcal{G}$, PRIVATE-SPARSIFY-ADD-MULT *outputs a Laplacian of a graph* $L_{\widetilde{\mathcal{G}}}$ *with the following guarantees:*

1. $L_{\widetilde{\mathcal{G}}}$ *is* $(\alpha, \beta)$-*differentially private with respect to the edge-level privacy.*

2. *With probability at least* $1 - \delta$, *we have*

$$O\left(\frac{w(1-\varepsilon)}{n}\log n\right)L_n - \varepsilon L_{\mathcal{G}} \preceq L_{\widetilde{\mathcal{G}}} \preceq O\left(\frac{w(1+\varepsilon)}{n}\log n\right)L_n + \varepsilon L_{\mathcal{G}}.$$

3. *The number of edges in* $L_{\widetilde{\mathcal{G}}}$ *is* $O(n/\varepsilon^2)$.

## E Why Traditional Approaches Do Not Work?

We argued briefly in the introduction that traditional approaches for spectral sparsification do not work. We also argued that traditional privacy mechanisms also fails to either give good spectral sparsification guarantees. In this section, we give the technical reasons why all these approaches do not work for privacy.

### E.1 Using Known Privacy Techniques

**Construction.** Compute the leverage score and then add noise scaled to its sensitivity. This would incur an error proportional to $O(\ell_{\mathsf{Lip}}n^2)$ if effective resistance has Lipschitz constant $\ell_{\mathsf{Lip}}$ as the vector of effective resistance has dimension $O(n^2)$. Unfortunately, we show that effective resistance is not Lipschitz continuous. In other words,

**Lemma 40.** *Effective resistance of edges is not a Lipschitz smooth function.*

*Proof.* Let $e_i$ be an edge in the graph $\mathcal{G}$ with weight 1. Let $\mathcal{G}'$ be a neighboring graph with all edges same as in $\mathcal{G}$ except for the edge $i$ that has weight 2. We can consider this action as a diagonal matrix $C$ acting on $E_\mathcal{G}$ where $C_{ii} = 2$ for $i \in [\binom{n}{2}]$ and $C_{jj} = 1$ for all $j \neq i$. In this notation $E_{\mathcal{G}'} = C E_\mathcal{G}$. Let denote by $\tau_i$ the effective resistance for an edge $i = (u, v)$ between nodes $u$ and $v$, i.e., $\tau_i := (E_\mathcal{G})_i (L_\mathcal{G})^\dagger (E_\mathcal{G})_i^\top$. Using [38, Corollary 3], we have

$$
\begin{aligned}
\tau_i' := \tau_i'(E_{\mathcal{G}'}) &= 2(E_\mathcal{G})_i \left( L_\mathcal{G} + 2(E_\mathcal{G})_i (E_\mathcal{G})_i^\top \right)^\dagger (E_\mathcal{G})_i^\top \\
&= 2(E_\mathcal{G})_i \left( L_\mathcal{G}^\dagger - 2 \frac{L_\mathcal{G}^\dagger (E_\mathcal{G})_i^\top (E_\mathcal{G})_i L_\mathcal{G}^\dagger}{1 + 2(E_\mathcal{G})_i L_\mathcal{G}^\dagger (E_\mathcal{G})_i^\top} \right) (E_\mathcal{G})_i^\top \\
&= 2 \left( \tau_i - 2 \times \frac{\tau_i^2}{1 + 2\tau_i} \right) \\
&= \frac{2\tau_i}{1 + 2\tau_i} \geq \frac{2}{3} \tau_i
\end{aligned}
$$

On the other hand, using [38, Corollary 3] again, we have for $j \neq i$,

$$
\begin{aligned}
\tau_j' := \tau_j'(E_{\mathcal{G}'}) &= (E_\mathcal{G})_j \left( L_\mathcal{G} + 2(E_\mathcal{G})_j (E_\mathcal{G})_j^\top \right)^\dagger (E_\mathcal{G})_j^\top \\
&= (E_\mathcal{G})_j \left( L_\mathcal{G}^\dagger - 2 \times \frac{L_\mathcal{G}^\dagger (E_\mathcal{G})_j^\top (E_\mathcal{G})_j L_\mathcal{G}^\dagger}{1 + 2(E_\mathcal{G})_j L_\mathcal{G}^\dagger (E_\mathcal{G})_j^\top} \right) (E_\mathcal{G})_j^\top \\
&= \tau_j - \frac{2\tau_j^2}{1 + 2\tau_j} \\
&= \tau_j
\end{aligned}
$$

In other words, effective resistances are not Lipschitz smooth. $\square$

**Objective Perturbation and Resampling.** Another option to preserve privacy is performing output perturbation. If we compute a sparse graph and then perform output perturbation, we need to perturb every possible edges. We can still sparsify this graph since differential privacy is preserved under post-processing; however, this procedure leads to an error term that scales proportional to $O(n^2)$. So any hope of using output perturbation seems to hit a road block.

**Recursive Sampling.** Another approach that one can try is to use recursive sampling, i.e., iterate the following few number of times: form a coarse sparsifier, add noise, and then sparsify again. This approach has seen success in the context of $k$-rank approximation where the general technique used is Krylov subspace iteration: compute a QR decomposition, then add noise, and then compute the QR decomposition again. Unfortunately, while in the case of low-rank approximation, the noise only scales proportional to $k$, in this case, the added noise would aggregate leading to an error term that scales proportional to $n$ and the number of iterations.

**Exponential Mechanism.** Note that the set of all sparsifiers is bounded above by $\exp(O(n \log(n)))$. Even though we managed to find a range of size $2^{O(n \log(n))}$, it is possible to show that the range of the mechanism has to be $2^{\Omega(n)}$. (Fix $\eta < 1/2$ and think of a set of inputs $\mathcal{G}$ where each graph has $n/2$ vertices with degree $n\eta$ and $n/2$ vertices with degree $n^{2\eta}$. Preserving all cuts of size 1 up to $(1 \pm \varepsilon)$ requires our output to have vertices of degree at least $(1 - \varepsilon)n^{2\eta}$ and vertices of degree less than $(1 + \varepsilon)n\eta$. Therefore, by representing vertices of high- and low-degree using a binary vector, there exists an injective mapping of balanced $\{0, 1\}^n$-vectors onto the set of potential outputs.) Thus, unless one can devise a scoring function of lower sensitivity, the exponential mechanism is bounded to have additive error proportional to $n/\alpha$.

## Footnotes

[5] A related problem is that of privately learning a robust subspace which was recently studied by [4].