[Reviews · NeurIPS 2019]

Reviewer 1



I have to preface my review by saying that I am not an expert on graphs and I am very unfamiliar with the results in this field. I will assess the paper based mostly on the privacy guarantees provided by the authors. The authors consider the problem of privately finding a sparse representation of a graph.That is find a graph with O(n) edges with a Laplacian close to that of the original graph. Their algorithm is based on computing effective resistances privately and then running a non private algorithm on these resistances to compute the sparsification. I find the motivation for the paper interesting. The proofs seemed to be correct and the algorithms are well explained. My only problem is that the algorithm appears to be intractable. Indeed the algorithm requires solving a SDP which could potentially very hard to solve. The fact that the authors provide no experiments or an implementation of their algorithm leads me to believe that this algorithm is in fact impractical. One minor clarification is: why do we need to sketch the left singular space to obtain the effective resistance? Can't we do that from the first sketch R?

Reviewer 2



The paper address privacy in graphs. The authors consider edge privacy and provide a mechanism for sparsification of a graph that ensures differential privacy and approximates the spectrum of the original graph. The sparse graph is then used for cuty queries. The authors use the measure of effective resistance in the probability for sampling an edge. It is still not clear to me that the sampling then does not divulge importance of an edge. Most of the technical discussion is brief and proofs are in the supplementary material. I would have hoped that a more intuitive explanation would make up for the proofs being in the supplementary material, but that is not the case here. There is no motivation relating to practical uses cases. Under what scenarios are private cut queries important? A discussion of the practical impact of privacy in the cases considered here would be helpful.

Reviewer 3



The paper is not motivated well. I agree that protecting privacy is important and that differential privacy is a strong and interesting guarantee, but the paper doesn't do a good job in convincing me that private graph sparsification is a relevant topic. How does this paper relate to actual machine learning applications? The authors claim that their approach provides practical algorithms for computing certain properties with better accuracies than previously possible. That's good, but I don't see any actual application and no evaluation of their approach (except for the abstract list in Table 1). While I appreciate the formal approach the authors took in building up their 41 (!) definitions, theorems and lemmas (to be fair, most of those are repetitions of existing formalism), I don't think that NeurIPS is the right venue for the paper. I found the description of their main algorithm (Algorithm 1) very cryptic and hard to follow too: I'm getting confused by the 6 or 7 different G's. We get G as an input, then we construct \hat G by mixing G with the so far undefined K_n (from the text, I think it's a fully connected graph). We create some L and that we only use to compute \tilde \tau_i's that we use to form a diagonal matrix (with some probabilities? Why?) D. We then construct another complete graph H with Gaussian edge weights, combine it with \hat G from earlier to get G'; then we need to solve this SDP-1 formula to get \bar G (explained somewhat in the text in the paragraph about G_int, whatever that is); somewhere in there we have a \tilde G' as well. Finally, we run some algorithm from [47] to get \tilde G. It's very much unclear to me how efficient all of these steps are, particularly solving SDP-1 over all potential graphs \bar G could be very inefficient. Update: I have read the author's response and it sadly did not help me me in understanding the paper and the relevance of its contributions. I apologize to the authors if my critically worded review or my lack of familiarity with their specific area of work has offended them -- merely citing the same lines I tried to decipher, however, does not make for a helpful response, even if done so angrily.

Reviewer 4



I am not an expert on the graph sparsification and I did not check the proof. I suggest the weak accept due to the following reasons: 1) The motivation of the problem is unclear. Although it is clear that the original graph sparsification is unsuitable to DP model. The relaxation problem is quite weird. Especially the spectral approximation condition. Why here they need to use a Laplacian of a complete graph, L_n here? What is the connection between this relaxation and the original problem? 2) Still, the motivation of the problem. I am not sure whether it is necessary to study the problem under edge DP. I think the authors should given an example why it is reasonable. 3) My reason of weak accept is due the approach in this paper is quite interesting. As the main idea is to calculate the effective resistance privately. I think it may could found other applications. ---------------------------After Rebuttal---------------------------------------------------- I changed to weak reject due to the following reasons. Definition 2 is quite weird. The authors should clarify why their definition is reasonable. Why here use Ln? I think the authors need to tell a complete story. Not just say something like it can be used to some problems. Since they just considered the edge DP, they need to say that Edge DP is reasonable in these problems. So I changed my opinion to weak reject.

[Author Response · NeurIPS 2019]

We thank all the reviewers for their comments.

**Reviewer 1**:

$R$ does not contain enough statistics to estimate effective resistances. Effective resistances of $m$ edges graph depends
on both left and right singular space: it is the diagonal entries of matrix $E_{\mathcal{G}}^{\top} L_{\mathcal{G}}^{\dagger} E_{\mathcal{G}}$. We can approximate $L_{\mathcal{G}}^{\dagger}$ by $R$, but
we need to also approximate the left singular space of $E_{\mathcal{G}}$ for the left and right multiplication in the above expression.

As for the complexity, the preprocessing time matches current state-of-the art result of Gupta et al. [28] as they also
require solving a quadratic program of Alon-Naor (since we can also answer $(S,T)$-cut queries, it is only fair to
compare with Gupta et al.; Blocki et al. only answer queries when $T = V \setminus S$). The total time to compute all $\tau_i$ for
$1 \leq i \leq m$ is $O(n^3)$. Solving a semi-definite program takes poly$(n)$ time, where the exact polynomial depends on
whether we use interior point, ellipsoid method, or primal-dual approach of Arora-Kale. Rest of all the computations is
subsumed by this run-time. However, now solving any $(S,T)$-cut query requires $O(\min |S|, |T|/\varepsilon^2)$ time instead of
$O(n^2)$ time required by Gupta et al. while achieving better accuracy bound. On top of that, since we now work with
sparse-graph, the run time of solving MAX-CUT and SPARSEST-CUT decreases significantly as the existing SDP
based algorithms have a large polynomial dependence on the number of edges.

**Reviewer 2**:

Note that edges that have high leverage score are more likely to be retained in the graph (this is necessary for the
utility/accuracy) but we also have plausible deniability for that edge, i.e., the edge could be in the output graph due
to the overlaid complete graph. How to balance the two is the subtle part of setting the appropriate parameters which
follows from analyzing the error bound. This is true for any differential privacy application, there is no absolute privacy.
We necessarily have to "leak" some information (in a controlled manner) to get some utility out of the analysis. We
discuss this briefly on lines 103-106, and lines 154-156.

Regarding the comment about " the tradeoff due to privacy, the privacy cost, cannot be understood in the current paper",
the whole point of giving the error bound is to crisply characterize that tradeoff. What we show (refer to Table 1) is that
if we require stronger privacy guarantee (by making privacy parameter $\alpha$ smaller), the upper bound on the error gets
worse (as $1/\alpha$). This is a typical tradeoff in applications of differential privacy.

**Reviewer 3**:

We do not think that the reviewer has even tried to read the paper as all of our response amounts to basically providing
pointers to the text in the paper. None of their comments support/justify the overly harsh evaluation. We urge the AC to
intervene and politely request to not consider the comments of the reviewer. Please see more details below.

Motivation/significance: We discuss a clear application to machine learning on lines 281-300. In particular, we discuss
how to extend our results for private manifold learning using Laplacian eigenmaps. More generally, as we argue in the
paper, graph analysis finds application in many problems in data science and machine learning. We focus on graph
sparsification as it is central to many graph analysis problems. This is clearly spelled out in the Introduction. More
precisely, the opening paragraph motivates the need for private analysis on graphs on lines 10-17, lists numerous
applications of graph analysis on lines 29-36, and finally discusses why graph sparsification plays a central role (on
lines 44-47 and lines 62-67).

Algorithm: There is a whole subsection (Section 3.1 on lines 129-163) that details each and every step of the algorithm,
and provides motivation and justification for each part. We find the comments by the reviewer as frivolous since answer
to each of their questions is easily accessible in that part of the paper. The algorithm is very clearly described in the text.

**General comment regarding empirical evaluation:**

This is an algorithms+theory paper. We give a general framework for privatizing analysis on graph. There is no
single application here, our results simultaneously apply to many problems. Besides as per the CFP, "Algorithmic
contributions should have at least an illustration of how the algorithm might eventually materialize into a machine
learning application." We give more than just illustrations, we give concrete applications as we discussed above and a
complete result for manifold learning. We disagree with the Reviewer 3 about the scope of NeurIPS.

More importantly, the applications are well established and studied, we would be reproducing old experiments without
adding any value. The point here is that the performance of the algorithms does not suffer much while guaranteeing
privacy. Establishing privacy empirically is not straightforward and therefore, many papers in the privacy track are not
accompanied by experiments.

[Meta-Review · NeurIPS 2019]

This paper presents a method for private graph sparsification under edge DP based on effective resistance sampling. This is an interesting contribution to the literature on DP with graph data which advances the state of the art. When preparing the final version of the paper, the authors must address the presentation issues raised by the reviewers, including: - A more concise description of the practical applications of their method, including a justification of why edge-level privacy is sufficient in such applications. - Properly introduce all the notation used in the manuscript. - A discussion to motivate Def 2, in particular the reasons behind the proposed notion of accuracy in point 3 as opposed to the one in Def 1. - A clear discussion of the computational cost of the algorithm to justify the efficiency claim in Thm 3.